# VT-FSL: Bridging Vision and Text with LLMs for Few-Shot Learning

**Wenhao Li[1,2], Qiangchang Wang[1]\*, Xianjing Meng[3], Zhibin Wu[1], Yilong Yin[1]\***

[1]School of Software, Shandong University    [2]Shenzhen Loop Area Institute
[3]School of Computing and Artificial Intelligence, Shandong University of Finance and Economics
{wenhao.li, zhibinwu}@mail.sdu.edu.cn,
rongmengyuan@gmail.com, {qiangchang.wang, ylyin}@sdu.edu.cn

## Abstract

Few-shot learning (FSL) aims to recognize novel concepts from only a few labeled support samples. Recent studies enhance support features by incorporating additional semantic information (e.g., class descriptions) or designing complex semantic fusion modules. However, these methods still suffer from hallucinating semantics that contradict the visual evidence due to the lack of grounding in actual instances, resulting in noisy guidance and costly corrections. To address these issues, we propose a novel framework, bridging Vision and Text with LLMs for Few-Shot Learning (VT-FSL), which constructs precise cross-modal prompts conditioned on Large Language Models (LLMs) and support images, seamlessly integrating them through a geometry-aware alignment mechanism. It mainly consists of Cross-modal Iterative Prompting (CIP) and Cross-modal Geometric Alignment (CGA). Specifically, the CIP conditions an LLM on both class names and support images to generate precise class descriptions iteratively in a single structured reasoning pass. These descriptions not only enrich the semantic understanding of novel classes but also enable the zero-shot synthesis of semantically consistent images. The descriptions and synthetic images act respectively as complementary textual and visual prompts, providing high-level class semantics and low-level intra-class diversity to compensate for limited support data. Furthermore, the CGA jointly aligns the fused textual, support, and synthetic visual representations by minimizing the kernelized volume of the 3-dimensional parallelotope they span. It captures global and nonlinear relationships among all representations, enabling structured and consistent multimodal integration. The proposed VT-FSL method establishes new state-of-the-art performance across ten diverse benchmarks, including standard, cross-domain, and fine-grained few-shot learning scenarios. Code is available at https://github.com/peacelwh/VT-FSL.

## 1 Introduction

Deep learning has achieved remarkable success in computer vision [1–3] and natural language processing [4] thanks to the availability of large-scale annotated datasets. However, collecting such data is often expensive or infeasible in many real-world scenarios [5–10]. Few-shot learning (FSL) [11] aims to address this challenge by enabling models to generalize from only a few labeled samples.

In FSL, the support set provides $N$ novel classes, each with $K$ labeled samples. The model learned from the support set is required to classify test samples from the query set. Among various approaches, metric-based methods [11–22] are widely adopted due to their superior scalability and performance.

---

\*Corresponding authors.

39th Conference on Neural Information Processing Systems (NeurIPS 2025).

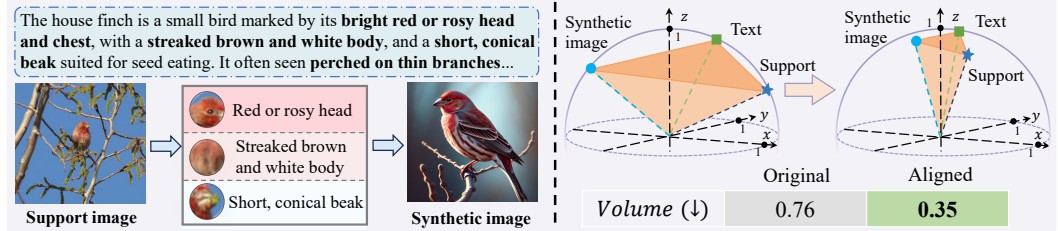

Figure 1: Illustration of the VT-FSL intuition. Left: The generated text and synthetic images provide high-level class semantics and low-level sample diversity. Right: By minimizing the volume of the 3-dimensional parallelotope spanned by all embeddings, they lie closer, indicating better alignment.

These methods embed both support and query samples into a shared feature space and classify each query by finding the nearest support sample. A common strategy is to construct class prototypes by averaging support features within each class to represent its semantic center [11]. However, the limited number of labeled examples makes it challenging to learn discriminative prototype representations, resulting in a semantic deviation from the true class center.

To address this issue, an alternative is to integrate additional semantic information from textual modality. Following this perspective, several studies [23–28] reveal that integrating semantics from class names can improve prototype representation. However, class names offer minimal contextual information. Several recent [29–31] works attempt to leverage Large Language Models (LLMs) [4] or external knowledge bases [32] to generate richer descriptions or attribute-based semantics to replace names. Despite their success, they rely solely on class names, neglecting valuable visual patterns of the support images. As a result, the generated text may lead to semantic hallucinations, i.e., misalignment with the actual corresponding object, which requires costly manual or algorithmic corrections. Moreover, naive input prompting hinders LLMs from fully utilizing the reasoning and generation capabilities, limiting the quality of semantics in few-shot scenarios.

In this paper, we propose a novel framework, bridging Vision and Text with LLMs for Few-Shot Learning (VT-FSL). First, a Cross-modal Iterative Prompting (CIP) module is introduced to condition LLMs [33, 34] on both class names and support images, obtaining precise and visually grounded descriptions. This is accomplished by a single structured inference pass, comprising four distinct stages for iteratively optimizing text quality: strategy, perception, refinement, and conclusion. Next, to enrich intra-class sample diversity, a text-to-image model [35, 36] is utilized to generate synthetic images with semantic consistency based on these descriptions in a zero-shot manner. As shown in Fig. 1 (left), the resulting text and images serve as complementary textual and visual prompts to compensate for limited support data. To fully utilize these cross-modal prompts, we incorporate textual embeddings from CLIP [37] into support features along spatial and channel dimensions via a lightweight two-layer perceptron. Furthermore, we propose a novel Cross-modal Geometric Alignment (CGA) that leverages the volume of a 3-dimensional parallelotope spanned by the fused support, textual, and synthetic visual embeddings for consistent alignment, as shown in Fig. 1 (right). CGA establishes a contrastive learning objective to enhance global and nonlinear relationships among all features by minimizing their volume in a kernelized parallelotope embedding space. Finally, VT-FSL achieves comprehensive cross-modal integration, enabling the extraction of generalized class prototypes enriched with discriminative information.

Overall, our contributions are summarized as follows:

- A VT-FSL framework is proposed to construct complementary cross-modal prompts with large language models, seamlessly integrating them through a geometry-aware alignment.

- A CIP module is proposed to generate precise descriptions conditioned on both class names and support images, driving zero-shot synthesis of semantically consistent examples.

- A CGA module is proposed to achieve comprehensive alignment across all representations and capture nonlinear semantic relations by kernelized volume-based contrastive learning.

- The proposed method achieves state-of-the-art performance on ten standard, cross-domain, and fine-grained FSL benchmarks, significantly improving accuracy by 4.2% on average.

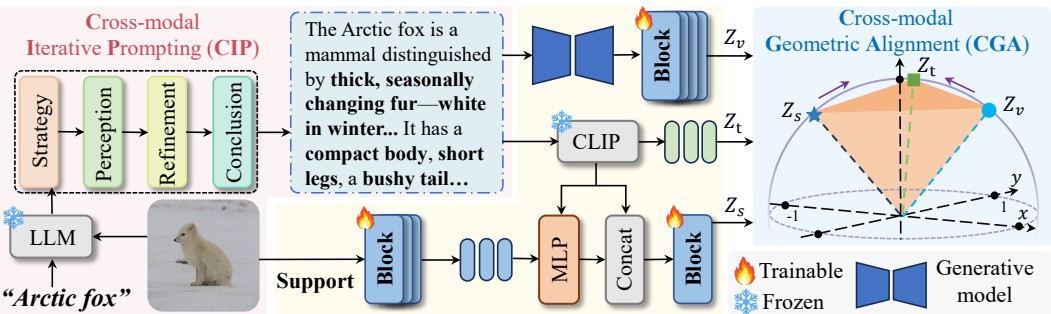

Figure 2: Overview of the proposed VT-FSL framework. First, given both class names and support images, CIP guides an LLM to generate precise descriptions via four structured stages. Synthetic images with semantic consistency are then generated based on these descriptions to expand the limited data. They are extracted to obtain features $Z_v$ by a feature extractor consisting of multiple Transformer blocks with shared weights. Next, the textual features $Z_t$ encoded by CLIP are injected into the support features $z_s$ via a two-layer MLP, enhancing the support embeddings $Z_s$. Finally, $Z_s$, $Z_t$, and $Z_v$ are jointly aligned through CGA, enabling global and nonlinear cross-modal interactions.

## 2  Related Work

**Few-shot Learning**. Existing few-shot learning (FSL) methods can be broadly categorized into two types. Optimization-based methods [38–41] adapt models to novel tasks with a few optimization steps, but may lead to meta-overfitting [42–44] due to the scarcity of task-specific supervision. In contrast, metric-based methods [11–22] learn a generalized embedding space where inter-class distances are maximized and intra-class distances are minimized. Several works also incorporate self-supervised learning to refine feature representations [20, 45–48]. To compensate for rare representative features within limited support data, many methods [23–28, 49] incorporate additional semantic information. For example, AM3 [25] fuses semantic features from class names with visual prototypes through an adaptive fusion mechanism. CaFo [28] generates synthetic images based on class names to expand the few-shot data. SIFT [49] generates high-quality features via a semantic transformation process. Several recent methods [29–31] explore the use of Large Language Models (LLMs) to replace class names with richer textual information. Conditioned on class names, ECER [30] extracts attribute-level textual information, and SemFew [31] constructs coherent descriptions to enhance prototype learning. In contrast, our approach jointly leverages both class names and support images to generate visually grounded textual descriptions via structured LLM reasoning and further produces semantically consistent synthetic images, forming complementary cross-modal prompts. Moreover, we fully integrate these prompts with support features in a geometry-aware alignment manner, capturing global and nonlinear cross-modal relationships.

**Contrastive Learning**. Most existing methods [50–55] adopt CLIP-based pairwise contrastive learning [37]. However, aligning each representation only to a single anchor neglects the interactions among the remaining points, making it difficult to capture global structural relationships and limiting the full integration of cross-modal semantics. The Gram matrix, which characterizes the mutual geometry among sets of vectors, has shown promise in theoretical analyses of deep learning networks [56] and various downstream tasks [57, 58]. To the best of our knowledge, we are the first to consider volume-based contrastive learning for global consistent alignment and further capture nonlinear relationships within a kernelized parallelotope embedding space in few-shot learning.

## 3  Methodology

We begin by introducing the preliminaries of FSL, and then present two components of our method: (1) explaining how textual and visual prompts are generated through CIP, and (2) how all embeddings are simultaneously aligned via CGA. Fig 2 shows the overview of the proposed VT-FSL.

### 3.1  Preliminaries

Few-Shot Learning (FSL) focuses on generalizing the knowledge learned from training set $C_{\text{train}}$ to test set $C_{\text{test}}$, where both sets are disjoint ($C_{\text{train}} \cap C_{\text{test}} = \emptyset$). FSL is typically formalized as an

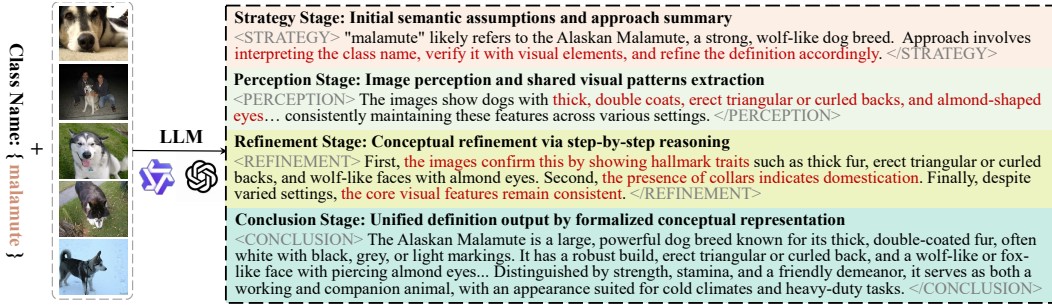

Figure 3: Illustration of Cross-modal Iterative Prompting (CIP). Given the class name and 5-shot support samples, CIP exploits the LLM through four structured reasoning stages: Strategy, Perception, Refinement, and Conclusion, to generate class-specific, precise class descriptions.

$N$-way $K$-shot classification task using an episodic strategy [12]. Each task consists of a support set $\mathcal{S} = \{(x_i, y_i)\}_{i=1}^{N \times K}$ and a query set $\mathcal{Q} = \{(x_i, y_i)\}_{i=1}^{N \times M}$ for performance evaluation.

In the FSL approach, prototype-based inference is commonly utilized, computing a prototype embedding $c_i$ for each class $i$ by averaging support features from the feature extractor $f_\phi$ as follows:

$$c_i = \frac{1}{K} \sum_{k=1}^{K} f_\phi(x_k), \quad x_k \in S_i \,. \tag{1}$$

For a query sample $\mathbf{x}^q$, the similarity score between $\mathbf{x}^q$ and all support classes is determined by calculating the distance function between $\mathbf{x}^q$ and $N$ class prototypes as follows:

$$p(y^q = i \mid \mathbf{x}^q) = \frac{\exp(cos(f_\phi(\mathbf{x}^q), c_i)/\tau)}{\sum_{j=1}^{N} \exp(cos(f_\phi(\mathbf{x}^q), c_j)/\tau)} \,, \tag{2}$$

where $\tau$ is a temperature parameter. $cos(\cdot, \cdot)$ denotes the cosine distance function. The support class with the maximum score is regarded as the classification result.

## 3.2 Cross-modal Iterative Prompting

To address the ambiguity in class names and capture the shared visual patterns from support samples, we propose Cross-modal Iterative Prompting (CIP), which jointly leverages the class label and $K$-shot support images to generate precise and visually grounded class descriptions. CIP follows a structured reasoning process inspired by Chain-of-Thought (CoT), which improves the semantic interpretability and generation accuracy of Large Language Models (LLMs).

CIP decomposes the generation process into four structured reasoning stages, including strategy, perception, refinement, and conclusion, as shown in Fig. 3. Every stage guided by the prompting design corresponds to outlining the problem, interpreting relevant information from the image, proceeding with a step-by-step reasoning process, and ultimately reaching a well-supported conclusion, respectively. Each stage is marked using structured tags (e.g., <STRATEGY>...</STRATEGY>), which allows the LLM to maintain reasoning boundaries throughout a single inference pass. This design removes the need for multi-turn interactions or external filtering, reducing manual effort and latency. The entire structural reasoning process and more cases are detailed in the Appendix.

As shown in Fig. 4, the generated description is then fed into a text-to-image generative model to produce synthetic images with semantic consistency in a zero-shot manner. To ensure semantic fidelity, we incorporate an LLM-based pairwise comparison strategy, which selects the top-$K$ images per class by ranking them against the textual description. An enriched support set is obtained, containing both real and synthetic samples without compromising the low-data regime. Formally, an $N$-way $(K + K)$-shot support set can be constructed from $N$ textual descriptions $T_N$ as:

$$\hat{S}_{N \times (K+K)} = \{\text{Synthesize}(T_N), \ S_{N \times K}\}. \tag{3}$$

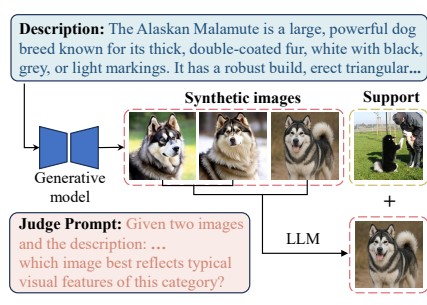

Figure 4: Illustration of visual synthetic images in the 1-shot task.

### 3.3 Cross-modal Geometric Alignment

**Cross-modal Fusion**. First, we leverage textual prompts to adaptively adjust support features extraction based on rich semantics. The textual features $Z_t$ encoded from the CLIP are linearly mapped to the same embedding space, followed by integrating with support features $z_s$ by a lightweight and effective two-layer network, as follows:

$$\beta = \sigma \left( W_2 \sigma \left( W_1 \left[ Z_t; \text{Avg}(z_s) \right] \right) \right), \tag{4}$$

where $W_i$ and $W_2$ are parameters of the network, $\sigma$ is sigmoid activation function. Avg( ) denotes the average of all patch tokens. The vector $\beta$ is injected into each token of support features, modulating support features along the channel dimension.

$Z_t$ and $z_s$ are then concatenated along the spatial dimension to capture the relevance between tokens based on rich semantics via Multi-head Self-Attention (MSA) of the Transformer block:

$$A = \text{Softmax} \left( \frac{QK^T}{\sqrt{d}} \right), \tag{5}$$

$$Z_s = (AV)W, \tag{6}$$

where $Q$, $K$, and $V$ are linearly mapped by each token. $d$ is the dimension of each head. The attention matrix $A$ is used to aggregate information from the value $V$. The outputs from all attention heads are concatenated and projected linearly by the weight matrix $W$, obtaining the enhanced $Z_s$.

**Kernelized Volume-based Contrastive Learning**. To achieve comprehensive consistency alignment, we propose a kernelized volume-based contrastive objective to improve global and nonlinear interactions among all features. Unlike traditional pairwise contrastive objectives, we measure the alignment among multiple vectors via the volume of the k-dimensional parallelotope they span in a shared embedding space [59]. A smaller volume indicates closer alignment. Specifically, $K$ normalized embeddings $\mathbf{v}_1, \ldots, \mathbf{v}_k \in \mathbb{R}^n$ construct the matrix $\mathbf{A} = [\mathbf{v}_1, \ldots, \mathbf{v}_k]$ and their extremities lie on the surface of the unit hypersphere. The Gram matrix $\mathbf{G}(\mathbf{v}_1, \ldots, \mathbf{v}_k) \in \mathbb{R}^{k \times k}$ is first defined, reflecting the square of the corresponding volume Vol as follows:

$$\mathbf{G}(\mathbf{v}_1, \ldots, \mathbf{v}_k) = \mathbf{A}^\top \mathbf{A} = \begin{bmatrix} \langle \mathbf{v}_1, \mathbf{v}_1 \rangle & \langle \mathbf{v}_1, \mathbf{v}_2 \rangle & \cdots & \langle \mathbf{v}_1, \mathbf{v}_k \rangle \\ \langle \mathbf{v}_2, \mathbf{v}_1 \rangle & \langle \mathbf{v}_2, \mathbf{v}_2 \rangle & \cdots & \langle \mathbf{v}_2, \mathbf{v}_k \rangle \\ \vdots & \vdots & \ddots & \vdots \\ \langle \mathbf{v}_k, \mathbf{v}_1 \rangle & \langle \mathbf{v}_k, \mathbf{v}_2 \rangle & \cdots & \langle \mathbf{v}_k, \mathbf{v}_k \rangle \end{bmatrix}. \tag{7}$$

$$\text{Vol}(\mathbf{v}_1, \ldots, \mathbf{v}_k) = \sqrt{\det(\mathbf{G})}, \quad \mathbf{G}_{ij} = \langle \mathbf{v}_i, \mathbf{v}_j \rangle. \tag{8}$$

To further model nonlinear interactions, this metric is extended into a high-dimensional Reproducing Kernel Hilbert Space (RKHS) using a Radial Basis Function (RBF) kernel mapping $\kappa(\cdot, \cdot)$. The volume is computed from the kernel Gram matrix $\mathbf{K}$:

$$\text{Vol}_{\mathcal{H}}(\mathbf{v}_1, \ldots, \mathbf{v}_k) = \sqrt{\det(\mathbf{K})}, \quad \mathbf{K}_{ij} = \kappa(\mathbf{v}_i, \mathbf{v}_j). \tag{9}$$

Next, Textual, synthetic, and enhanced support features are transformed into normalized triplets $(\mathbf{Z}_t, \mathbf{Z}_v, \mathbf{Z}_s)$. Selecting textual modality as the anchor focuses the alignment around it. Formal derivations of relevant theory and ablations on anchor and kernel function choices are detailed in the Appendix. Minimizing the kernelized volume via the contrastive loss is defined as:

$$\mathcal{L}_{\text{D2A}} = -\frac{1}{B} \sum_{i=1}^{B} \log \frac{\exp \left( -\text{Vol}_{\mathcal{H}}(Z_t^i, Z_s^i, Z_v^i)/\tau \right)}{\sum_{j=1}^{K} \exp \left( -\text{Vol}_{\mathcal{H}}(Z_t^j, Z_s^i, Z_v^i)/\tau \right)}, \tag{10}$$

$$\mathcal{L}_{\text{A2D}} = -\frac{1}{B} \sum_{i=1}^{B} \log \frac{\exp \left( -\text{Vol}_{\mathcal{H}}(Z_t^i, Z_s^i, Z_v^i)/\tau \right)}{\sum_{j=1}^{K} \exp \left( -\text{Vol}_{\mathcal{H}}(Z_t^i, Z_s^j, Z_v^j)/\tau \right)}. \tag{11}$$

Finally, the overall loss also includes the Cross-Entropy loss between the probability $p_i$ of the query sample $q$ to the $i$-th class in Eq. (2) and the corresponding ground-truth label, as follows:

$$\mathcal{L}_{\text{total}} = \sum_{i=1}^{M} \text{CrossEntropy}(p_i, y_i) + \frac{1}{2}(\mathcal{L}_{\text{D2A}} + \mathcal{L}_{\text{A2D}}). \tag{12}$$

Table 1: Results (%) on *mini*ImageNet [12] and *tiered*ImageNet [60]. The average accuracy with 95% confidence interval is reported. Bold and blue font indicates the best and suboptimal results.

| Model | Venue | Backbone | $\approx$ # Params | *mini*ImageNet | | *tiered*ImageNet | |
|---|---|---|---|---|---|---|---|
| | | | | 1-shot | 5-shot | 1-shot | 5-shot |
| MatchNet [12] | NeurIPS'16 | ResNet-12 | 12.4M | $65.64_{\pm0.20}$ | $78.72_{\pm0.15}$ | $68.50_{\pm0.92}$ | $80.60_{\pm0.71}$ |
| ProtoNet [11] | NeurIPS'17 | ResNet-12 | 12.4M | $62.39_{\pm0.21}$ | $80.53_{\pm0.14}$ | $68.23_{\pm0.23}$ | $84.03_{\pm0.16}$ |
| AM3 [25] | NeurIPS'19 | ResNet-12 | 12.4M | $65.30_{\pm0.49}$ | $78.10_{\pm0.36}$ | $69.08_{\pm0.47}$ | $82.58_{\pm0.31}$ |
| DeepEMD [14] | CVPR'20 | ResNet-12 | 12.4M | $65.91_{\pm0.82}$ | $82.41_{\pm0.56}$ | $71.16_{\pm0.87}$ | $86.03_{\pm0.58}$ |
| PCPK [29] | CVPR'21 | ResNet-12 | 12.4M | $73.13_{\pm0.85}$ | $82.06_{\pm0.54}$ | $81.04_{\pm0.89}$ | $87.42_{\pm0.57}$ |
| SUN [17] | ECCV'22 | Visformer-S | 12.3M | $67.80_{\pm0.45}$ | $83.25_{\pm0.30}$ | $72.99_{\pm0.50}$ | $86.74_{\pm0.33}$ |
| FewTURE [47] | NeurIPS'22 | ViT-S/16 | 22.0M | $68.02_{\pm0.88}$ | $84.51_{\pm0.53}$ | $72.96_{\pm0.92}$ | $86.43_{\pm0.67}$ |
| SVAE [27] | CVPR'22 | ResNet-12 | 12.4M | $74.84_{\pm0.23}$ | $83.28_{\pm0.40}$ | $76.98_{\pm0.65}$ | $85.77_{\pm0.50}$ |
| Meta-AdaM [61] | NeurIPS'23 | ResNet-12 | 12.4M | $59.89_{\pm0.49}$ | $77.92_{\pm0.43}$ | $65.31_{\pm0.48}$ | $85.24_{\pm0.35}$ |
| ProtoDiff [62] | NeurIPS'23 | ResNet-12 | 12.4M | $66.63_{\pm0.21}$ | $83.48_{\pm0.15}$ | $72.95_{\pm0.24}$ | $85.15_{\pm0.18}$ |
| CPEA [63] | ICCV'23 | ViT-S/16 | 22.0M | $71.97_{\pm0.65}$ | $87.06_{\pm0.38}$ | $76.93_{\pm0.70}$ | $90.12_{\pm0.45}$ |
| SP [23] | CVPR'23 | Visformer-T | 10.0M | $72.31_{\pm0.40}$ | $83.42_{\pm0.30}$ | $78.03_{\pm0.46}$ | $88.55_{\pm0.32}$ |
| NIW-Meta [64] | ICLR'24 | WRN28-10 | 36.5M | $68.54_{\pm0.26}$ | $84.81_{\pm0.28}$ | $74.59_{\pm0.33}$ | $89.76_{\pm0.23}$ |
| FeatWalk [16] | AAAI'24 | ResNet-12 | 12.4M | $70.21_{\pm0.44}$ | $87.38_{\pm0.27}$ | $75.25_{\pm0.48}$ | $89.92_{\pm0.29}$ |
| BECLR [65] | ICLR'24 | ResNet-18 | 11.7M | $75.74_{\pm0.62}$ | $84.93_{\pm0.33}$ | $76.44_{\pm0.66}$ | $84.85_{\pm0.37}$ |
| SIFT [49] | IJCV'24 | WRN-28-10 | 36.5M | $77.31_{\pm0.67}$ | $86.95_{\pm0.53}$ | $77.86_{\pm0.77}$ | $89.89_{\pm0.52}$ |
| SemFew [31] | CVPR'24 | Swin-T | 29.0M | $78.94_{\pm0.66}$ | $86.49_{\pm0.50}$ | $82.37_{\pm0.77}$ | $89.89_{\pm0.52}$ |
| UAP [66] | NeurIPS'24 | ResNet-12 | 12.4M | $81.63_{\pm0.28}$ | $79.05_{\pm0.19}$ | $79.68_{\pm0.30}$ | $76.78_{\pm0.21}$ |
| ECER [30] | AAAI'25 | Visformer-T | 10.0M | $81.14_{\pm0.15}$ | - | $81.81_{\pm0.51}$ | - |
| VT-FSL | ours | Visformer-T | 10.0M | $\mathbf{83.66_{\pm0.31}}$ | $\mathbf{88.38_{\pm0.25}}$ | $\mathbf{88.02_{\pm0.34}}$ | $\mathbf{91.71_{\pm0.27}}$ |

We detail the algorithm of the entire training process in the Appendix. During testing, the prototype with textual prompts $c_t$ is obtained by averaging the enhanced support features. The prototype with visual prompts $c_v$ is computed by averaging the expanding support set in Eq. (3). The final classification prototype $C$ is then obtained by integrating $c_v$ and $c_t$ in a convex combination manner [25]:

$$C = uc_t + (1 - u)c_v, \tag{13}$$

where $u \in [0, 1]$ is a manually controlled fusion factor, determined in the validation set.

## 4 Experiments

### 4.1 Experimental Details

**Datasets**. Extensive experiments are conducted in three distinct few-shot learning scenarios. (1) Four datasets in standard FSL: *mini*ImageNet [12], *tiered*TieredNet [60], CIFAR-FS [67], and FC100 [19]. (2)Three datasets in fine-grained FSL: CUB [68], Cars [69], and Dogs [70]. (3)Three datasets in cross-domain FSL: CUB, Places [71], and Plantae [72]. The detailed dataset statistics regarding the number of categories and images are introduced in the Appendix.

**Implementation Details**. Following recent few-shot studies [24, 23, 30], we adopt Visformer-Tiny [3] as the feature extractor and apply the text encoder from ViT-B/16 CLIP [37] where the output dimension is 512. Qwen2.5-VL-32B [34] and Janus-Pro [36] models are utilized to generate textual and visual prompts by default. More architecture comparisons are detailed in the Appendix. Our training framework adopts a two-stage framework [17], consisting of pre-training and meta-tuning stages. Input images are resized into 224×224 [31]. The AdamW optimizer [73] is used with a learning rate of 5e-4 and a cosine scheduler. Pre-training runs for 300 epochs in *tiered*ImageNet and 800 epochs in other datasets with a batch size of 512, followed by meta-tuning for 100 epochs via an episodic training strategy. The hyperparameter $\tau$ is set as 0.2 according to validation accuracy. All experiments are performed with an NVIDIA RTX 6000 Ada.

**Evaluation Protocol**. For evaluation, we adopt the widely used episodic protocol [23–28] in FSL. Specifically, 2000 classification tasks are uniformly sampled from the novel classes that do not overlap with training categories. Each task follows the standard $N$-way $K$-shot setting, where 15 query samples per class are included for evaluation. The final performance is reported as the mean classification accuracy across all sampled tasks, along with the 95% confidence interval.

### 4.2 Main Results

**Standard Few-Shot Classification**. Table 1 and Table 2 show the comparative results on four benchmarks. It is worth noting that (1) Semantic-based methods (AM3, PCPK, SVAE, SP, SIFT,

SemFew, and ECER) generally outperform others, highlighting the effectiveness of semantic information in learning more generalizable feature representations. (2) The proposed VT-FSL uses a lightweight Visformer-T backbone yet outperforms all methods with larger backbones (e.g., ViT-S/16, Swin-T, and WRN28-10), demonstrating that VT-FSL fully utilizes the feature extraction potential of the backbone. (3) VT-FSL outperforms previous methods by a large margin. Compared with the competing methods on four benchmarks, VT-FSL surpasses SemFew [31]

Table 2: Results (%) with the average accuracy is reported on CIFAR-FS [67] and FC100 [19].

| Method | Venue | CIFAR-FS | | FC100 | |
|---|---|---|---|---|---|
| | | 1-shot | 5-shot | 1-shot | 5-shot |
| ProtoNet [11] | NeurIPS'17 | 72.20 | 83.50 | 41.54 | 57.08 |
| MABAS [74] | ECCV'20 | 73.51 | 85.65 | 42.31 | 58.16 |
| FewTURE [47] | NeurIPS'22 | 77.76 | 88.90 | 47.68 | 63.81 |
| MAdaM [61] | NeurIPS'23 | – | – | 41.12 | 56.14 |
| SP [23] | CVPR'23 | 82.18 | 88.24 | 48.53 | 61.55 |
| ALFA [39] | TPAMI'24 | 76.32 | 86.73 | 44.54 | 58.44 |
| LastShot [40] | TPAMI'24 | 76.76 | 87.49 | 44.08 | 59.14 |
| SemFew [31] | CVPR'24 | 84.34 | 89.11 | 54.27 | 65.02 |
| ECER [30] | AAAI'25 | 86.01 | – | 57.34 | – |
| VT-FSL | ours | 88.67 | 91.45 | 57.99 | 67.68 |

by 3.7%-5.7% and 1.0%-2.7% in the 1-shot and 5-shot settings. Moreover, compared to complex fusion mechanisms [25–27], VT-FSL achieves such improvements using a simple two-layer network with only 0.7M parameters, significantly fewer than the 4.3M parameters used in SemFew. These results highlight the effectiveness of constructing complementary multimodal prompts with global alignment across all features for full semantic integration.

Table 3: Results (%) on CUB [68], Dogs [70], and Cars [69]. The average accuracy with 95% confidence interval is reported. Bold and blue font indicates the best and suboptimal results.

| Method | Venue | CUB-200-2011 | | Stanford-Dogs | | Stanford-Cars | |
|---|---|---|---|---|---|---|---|
| | | 1-shot | 5-shot | 1-shot | 5-shot | 1-shot | 5-shot |
| ProtoNet [62] | NeurIPS'17 | $63.44_{\pm0.56}$ | $83.17_{\pm0.35}$ | $41.61_{\pm0.50}$ | $76.78_{\pm0.36}$ | $45.01_{\pm0.49}$ | $87.19_{\pm0.31}$ |
| FRN [75] | CVPR'21 | $83.55_{\pm0.19}$ | $92.92_{\pm0.10}$ | $49.37_{\pm0.20}$ | $67.13_{\pm0.17}$ | $58.90_{\pm0.22}$ | $79.65_{\pm0.15}$ |
| DAN [76] | AAAI'22 | $72.89_{\pm0.50}$ | $86.60_{\pm0.31}$ | $59.81_{\pm0.50}$ | $77.19_{\pm0.35}$ | $70.21_{\pm0.50}$ | $85.55_{\pm0.31}$ |
| MFGN [77] | IJCAI'22 | $84.01_{\pm0.39}$ | $84.01_{\pm0.39}$ | $74.81_{\pm0.44}$ | $86.52_{\pm0.26}$ | - | - |
| TDM [78] | CVPR'22 | $84.36_{\pm0.19}$ | $93.37_{\pm0.10}$ | $57.64_{\pm0.22}$ | $75.03_{\pm0.16}$ | $68.36_{\pm0.22}$ | $86.14_{\pm0.13}$ |
| BSFA [79] | TCSVT'23 | $86.00_{\pm0.41}$ | $92.53_{\pm0.23}$ | $69.58_{\pm0.50}$ | $82.59_{\pm0.33}$ | $88.93_{\pm0.38}$ | $95.20_{\pm0.20}$ |
| MLI [80] | TIP'24 | $85.94_{\pm0.42}$ | $93.50_{\pm0.29}$ | $76.32_{\pm0.47}$ | $88.25_{\pm0.27}$ | - | - |
| C2-Net [81] | AAAI'24 | $83.31_{\pm0.41}$ | $92.18_{\pm0.23}$ | $75.50_{\pm0.49}$ | $87.65_{\pm0.28}$ | $88.96_{\pm0.37}$ | $95.16_{\pm0.20}$ |
| SUITED [82] | AAAI'25 | $86.02_{\pm0.47}$ | $94.13_{\pm0.24}$ | $76.55_{\pm0.47}$ | $88.86_{\pm0.27}$ | $89.97_{\pm0.36}$ | $96.53_{\pm0.16}$ |
| VT-FSL | ours | $91.08_{\pm0.28}$ | $94.63_{\pm0.19}$ | $86.58_{\pm0.30}$ | $90.69_{\pm0.25}$ | $92.95_{\pm0.24}$ | $96.62_{\pm0.15}$ |

**Fine-grained Few-Shot Classification**. The classification results on three fine-grained datasets are presented in Table 3. It can be observed that the proposed VT-FSL method obtains the best classification results, outperforming the second-best SUITED [82] method by a significant margin of 3.0%-10.3% in the challenging 1-shot task across three benchmarks, and is far superior to other competing methods. This shows that our VT-FSL can also be effective on the fine-grained few-shot image classification tasks to capture subtle inter-class differences and preserve intra-class consistency by bridging the cross-modal semantic gap and enhancing multimodal integration.

Table 4: The average accuracy (%) is reported on cross-domain *mini*ImageNet [12] →CUB [68], Places [68], and Plantae [72]. Bold and blue font indicates the best and suboptimal results.

| Method | Venue | CUB | | Places | | Plantae | |
|---|---|---|---|---|---|---|---|
| | | 1-shot | 5-shot | 1-shot | 5-shot | 1-shot | 5-shot |
| GNN [83] | ICLR'18 | $44.40_{\pm0.68}$ | $62.87_{\pm0.65}$ | $52.42_{\pm0.80}$ | $70.91_{\pm0.65}$ | $33.60_{\pm0.56}$ | $48.51_{\pm0.59}$ |
| FWT [84] | ICLR'20 | $45.50_{\pm0.75}$ | $64.97_{\pm0.68}$ | $53.44_{\pm0.79}$ | $70.70_{\pm0.67}$ | $32.56_{\pm0.58}$ | $49.66_{\pm0.62}$ |
| AFA [85] | ECCV'22 | $46.86_{\pm0.70}$ | $68.25_{\pm0.65}$ | $54.04_{\pm0.75}$ | $76.21_{\pm0.60}$ | $36.76_{\pm0.65}$ | $54.26_{\pm0.68}$ |
| UCD [86] | NeurIPS'22 | $40.65_{\pm0.68}$ | $58.54_{\pm0.70}$ | $51.84_{\pm0.72}$ | $72.19_{\pm0.60}$ | $37.28_{\pm0.67}$ | $54.15_{\pm0.66}$ |
| ATA [87] | AIJ'23 | $45.00_{\pm0.50}$ | $66.22_{\pm0.50}$ | $53.57_{\pm0.50}$ | $75.48_{\pm0.40}$ | $34.42_{\pm0.40}$ | $52.69_{\pm0.40}$ |
| LDP-net [88] | CVPR'23 | $49.82_{\pm0.70}$ | $70.39_{\pm0.66}$ | $53.82_{\pm0.71}$ | $72.90_{\pm0.63}$ | $39.84_{\pm0.68}$ | $58.49_{\pm0.69}$ |
| StyleAdv [89] | CVPR'23 | $48.49_{\pm0.72}$ | $68.72_{\pm0.67}$ | $58.58_{\pm0.83}$ | $77.73_{\pm0.62}$ | $41.13_{\pm0.67}$ | $61.52_{\pm0.68}$ |
| FAP [90] | IJCAI'24 | $50.56_{\pm0.73}$ | $64.17_{\pm0.69}$ | $57.34_{\pm0.72}$ | $72.05_{\pm0.60}$ | $37.44_{\pm0.64}$ | $53.58_{\pm0.66}$ |
| FLoR [91] | CVPR'24 | $49.99_{\pm0.68}$ | $70.39_{\pm0.67}$ | $53.18_{\pm0.70}$ | $72.31_{\pm0.62}$ | $40.10_{\pm0.65}$ | $55.80_{\pm0.66}$ |
| MEFP [92] | NeurIPS'24 | $51.55_{\pm0.70}$ | $73.61_{\pm0.66}$ | $52.06_{\pm0.69}$ | $73.78_{\pm0.61}$ | $41.55_{\pm0.65}$ | $61.39_{\pm0.67}$ |
| SVasP [93] | AAAI'25 | $49.49_{\pm0.72}$ | $68.95_{\pm0.66}$ | $59.07_{\pm0.81}$ | $77.78_{\pm0.62}$ | $41.22_{\pm0.62}$ | $60.63_{\pm0.64}$ |
| VT-FSL | ours | $66.86_{\pm0.47}$ | $81.02_{\pm0.36}$ | $73.68_{\pm0.41}$ | $81.52_{\pm0.33}$ | $45.90_{\pm0.40}$ | $61.54_{\pm0.38}$ |

**Cross-Domain Few-Shot Classification.** Following the setup [85, 87], the model is trained on the training set of *mini*ImageNet and evaluated on three novel datasets: *mini*ImageNet → CUB [68], Places [68], and Plantae [72], which is more challenging due to significant domain shifts. As shown in

Table 5: Ablation study on three datasets under the 1-shot and 5-shot settings. $P_{text}$ means textual prompts, and $P_{vision}$ means visual prompts. $\mathcal{L}_{align}$ is kernelized volume-based contrastive loss.

| $P_{text}$ | $P_{vision}$ | $\mathcal{L}_{align}$ | miniImageNet | | CIFAR-FS | | tieredImageNet | |
|---|---|---|---|---|---|---|---|---|
| | | | 1-shot | 5-shot | 1-shot | 5-shot | 1-shot | 5-shot |
| | | | $68.47_{\pm0.43}$ | $82.63_{\pm0.30}$ | $76.43_{\pm0.45}$ | $86.60_{\pm0.31}$ | $75.88_{\pm0.37}$ | $87.39_{\pm0.29}$ |
| ✓ | | | $78.82_{\pm0.36}$ | $86.01_{\pm0.27}$ | $84.76_{\pm0.36}$ | $89.23_{\pm0.29}$ | $86.15_{\pm0.38}$ | $89.65_{\pm0.31}$ |
| ✓ | | ✓ | $78.96_{\pm0.36}$ | $86.35_{\pm0.29}$ | $86.54_{\pm0.34}$ | $89.74_{\pm0.29}$ | $86.91_{\pm0.37}$ | $90.19_{\pm0.29}$ |
| | ✓ | | $79.17_{\pm0.35}$ | $86.26_{\pm0.28}$ | $86.01_{\pm0.34}$ | $89.75_{\pm0.28}$ | $85.54_{\pm0.38}$ | $90.01_{\pm0.29}$ |
| | ✓ | ✓ | $79.76_{\pm0.35}$ | $86.72_{\pm0.28}$ | $86.64_{\pm0.34}$ | $90.13_{\pm0.28}$ | $85.93_{\pm0.38}$ | $90.30_{\pm0.29}$ |
| ✓ | ✓ | | $82.08_{\pm0.31}$ | $87.06_{\pm0.27}$ | $87.72_{\pm0.33}$ | $90.14_{\pm0.29}$ | $87.13_{\pm0.36}$ | $90.78_{\pm0.28}$ |
| ✓ | ✓ | ✓ | $\mathbf{83.66_{\pm0.31}}$ | $\mathbf{88.38_{\pm0.25}}$ | $\mathbf{88.67_{\pm0.32}}$ | $\mathbf{91.45_{\pm0.28}}$ | $\mathbf{88.02_{\pm0.34}}$ | $\mathbf{91.71_{\pm0.27}}$ |

Table 4, the proposed VT-FSL consistently outperforms all baselines by a large margin. Specifically, in the 1-shot task, VT-FSL surpasses the second-best methods by 4.35% (vs. SVasP) to 15.31% (vs. MEFP) across the three datasets. These results demonstrate that VT-FSL effectively learns more transferable representations by establishing a comprehensive interaction with the class-specific cross-modal semantics, generalizing well to novel categories even under distribution shifts.

### 4.3 Model Analysis

**Ablation Study**. We conduct an ablation study on three datasets to evaluate the effectiveness of our VT-FSL, as shown in Table 5. It is to be noted that (1) Introducing textual prompts alone improves performance over the baseline across all datasets, showing that precise and visually grounded semantics facilitate the extraction of discriminative features. Adding the kernelized volume-based contrastive loss, i.e., alignment loss, further improves results, especially on miniImageNet and tieredImageNet 1-shot settings, suggesting better cross-modal consistency. (2) Visual prompts also bring performance gains. The model with only visual prompts outperforms the baseline, and combining them with the alignment loss achieves further improvements from 89.75% to 90.14% in the 5-shot task on CIFAR-FS, indicating the alignment loss enhances intra-modal structure. (3) The best performance is achieved when both prompts and alignment loss are used. The full model consistently outperforms all ablations, demonstrating the complementarity of textual and visual prompts and the importance of jointly aligning them with the proposed geometric objective.

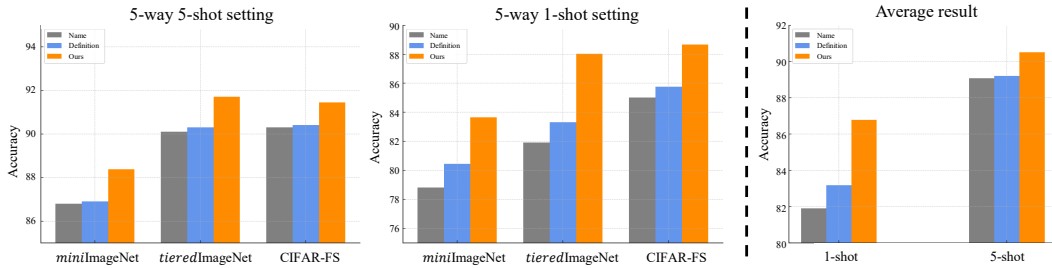

Figure 5: Comparison with textual semantics from name, definition of SemFew [31], and ours.

**Comparison with Different Textual Semantics**. As shown in Fig. 5, the texts generated by the proposed VT-FSL method achieve the best performance across all settings on three datasets, followed by SemFew [31, 30, 31] definitions and class names. This trend suggests that class names provide minimal information and that SemFew definitions, although generated by the LLM, rely only on class names and naive prompting, limiting the effectiveness. VT-FSL yields richer and more precise textual semantics by combining class names with support images and structured iterative prompting.

**Comparison with Contrastive Learning Methods**. As shown in Table 6a, we compare three contrastive learning strategies: InfoNCE [94], volume-based loss [58], and our proposed kernelized volume-based loss. The baseline is utilized without contrastive objective. InfoNCE performs the worst, especially in the 1-shot setting, due to its reliance on pairwise similarity, which overlooks interactions among multiple modalities. While the volume-based loss improves upon this, it remains limited to linear space and fails to model complex semantic structures. In contrast, our method

Table 6: Comparison with contrastive learning methods and different prototypes for inference

(a) Contrasting learning methods

| Method | *mini*ImageNet | | CIFAR-FS | |
|---|---|---|---|---|
| | 1-shot | 5-shot | 1-shot | 5-shot |
| baseline | 82.49 | 87.35 | 87.01 | 90.34 |
| InfoNCE [94] | 79.96 | 86.61 | 86.47 | 88.35 |
| Volume [58] | 82.33 | 87.59 | 87.20 | 89.72 |
| VT-FSL | **83.66** | **88.38** | **88.67** | **91.45** |

(b) Inference prototypes

| Method | *mini*ImageNet | | CIFAR-FS | |
|---|---|---|---|---|
| | 1-shot | 5-shot | 1-shot | 5-shot |
| $C$ | 71.63 | 86.13 | 80.75 | 90.34 |
| $T \Rightarrow C$ | 80.05 | 86.92 | 87.43 | 90.43 |
| $V \Rightarrow C$ | 82.83 | 87.55 | 87.27 | 90.63 |
| $T + V \Rightarrow C$ | **83.66** | **88.38** | **88.67** | **91.45** |

introduces a Reproducing Kernel Hilbert Space (RKHS) to enable nonlinear alignment, substantially enhances the ability to integrate cross-modal semantics and achieves the best results across settings.

**Comparison with Different Inference Prototypes**. In Table 6b, using only support features ($C$) yields the lowest accuracy, indicating limited discriminative power under scarce supervision. Incorporating textual or visual prompts significantly improves performance, demonstrating the value of external semantic guidance. The best results are obtained when both prompts are jointly integrated ($T + V \Rightarrow C$), confirming their complementarity and the effectiveness of cross-modal synergy.

Table 7: Comparison of training and inference times on *mini*ImageNet under 5-way 1-shot tasks.

| Method | Prompt (h) | Training (min) | Inference (ms) | Acc (%) |
|---|---|---|---|---|
| SP [23] | - | 1.7 | 78 | $72.31_{\pm 0.40}$ |
| SemFew [31] | 1.5 | 2.3 | 105 | $78.94_{\pm 0.66}$ |
| ECER [30] | 0.7 | 3.0 | 119 | $81.14_{\pm 0.15}$ |
| **VT-FSL (ours)** | **0.7** | **1.1** | **76** | $\mathbf{83.66_{\pm 0.31}}$ |

**The effect of computational overhead with LLMs**. To further investigate the efficiency of VT-FSL, we analyze the impact of computational overhead introduced by large language models (LLMs). The experiment is conducted on the same server configuration, comparing to SP [23] without any large models to generate descriptions/images and SemFew [31] and ECER [30] with these models. SP and ECER employ the same Visformer-T backbone as VT-FSL during training/inference. As shown in the Table 7, VT-FSL achieves both the lowest training time and inference time, while attaining the highest accuracy (83.66%). Compared to ECER, VT-FSL reduces training time from 3.0 min to 1.1 min and inference time from 119 ms to 76 ms, while improving accuracy by 2.5%. These results confirm that the plug-in design of VT-FSL transfers knowledge from large models efficiently in a one-time offline stage to construct cross-modal prompts. Any downstream FSL model can directly use the resulting cross-modal prompts, adding virtually no extra overhead during training or inference. This avoids the extra manual or algorithmic corrections required by SemFew and ECER, leading to a better trade-off between efficiency and performance.

**The effect of fusion factor** $u$. $u$ controls the relative weight between textual and visual features to obtain the final inference prototypes in Eq. 13. For each dataset, $u$ is automatically determined by a grid search over $[0, 1]$ on the corresponding validation set, selecting the value that yields the highest validation accuracy. As shown in the Fig. 6 , The optimal values are 0.5 (miniImageNet), 0.7 (tieredImageNet), 0.6 (CIFAR-FS), and 0.6 (FC100). The results show that performance peaks at intermediate values of $u$, confirming that textual and visual features are complementary. Using only visual prompts ($u = 0$) or only textual prompts ($u = 1$) consistently leads to inferior results.

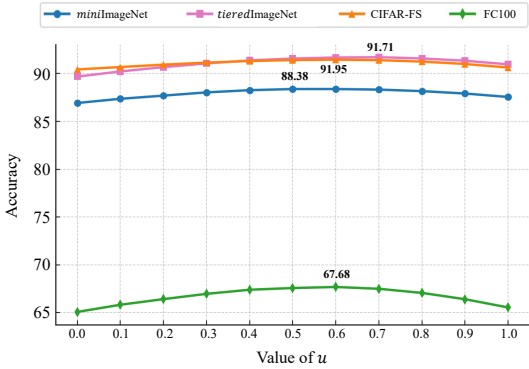

Figure 6: The effect of the fusion factor $u$.

**The effect of the Synthetic Image Number**. We evaluate how the number of synthetic images per class affects performance in the 1-shot setting, as shown in Fig. 7a. Accuracy improves notably when

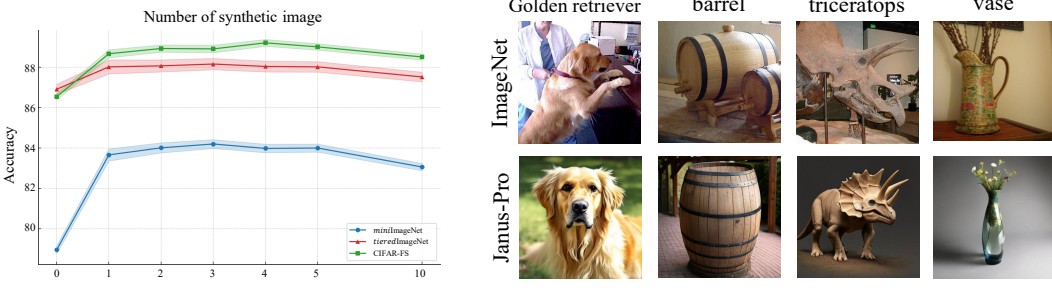

(a) The effect of the generated number

(b) Visualization of generated images

Figure 7: The effect of the generated number and visualization of generated images.

adding 1-3 synthetic images, but plateaus or slightly declines beyond that. This trend is consistent across all datasets. These results validate our design choice of generating $K$ synthetic images for $K$-shot tasks, which effectively augments the support set without compromising the low-data regime. However, increasing the number further introduces noise, as lower-ranked generations tend to be less discriminative and may degrade representation quality. In Fig 7b, we visualize the synthetic images generated by Janus-Pro on ImageNet [95]. Benefiting from precise descriptions with key visual attributes, synthetic images can well highlight the low-level semantics of the target category.

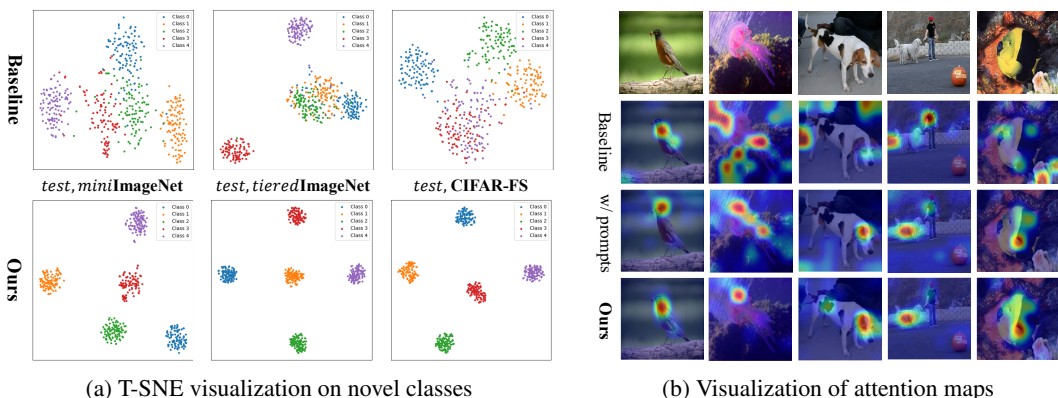

(a) T-SNE visualization on novel classes

(b) Visualization of attention maps

Figure 8: Visualization of T-SNE and attention maps.

**Visualization of T-SNE and Attention Maps**. Fig. 8a shows that VT-FSL produces well-separated and compact class clusters, in contrast to the scattered distributions of the baseline, indicating improved semantic alignment. As shown in Fig. 8b, the baseline suffers from attention to irrelevant regions, while our method focuses more accurately on key object areas, guided by cross-modal prompts and further enhanced through geometry-aware semantic alignment.

## 5    Conclusion

In this paper, we propose VT-FSL, a novel framework that bridges vision and text representations with large language models to advance few-shot learning by generating complementary cross-modal prompts and integrating them via a geometry-aware consistency alignment. Specifically, we introduce Cross-modal Iterative Prompting (CIP) to generate precise descriptions with high-level semantics from both class names and support images by a single structure inference pass, enabling the zero-shot synthesis of semantically consistent images with low-level diversity. We further propose Cross-modal Geometric Alignment (CGA) to comprehensively align the fused textual, support, and synthetic features by minimizing the volume in their kernelized parallelotope space, capturing global and nonlinear cross-modal dependencies. Extensive experiments on ten FSL benchmarks demonstrate the effectiveness of VT-FSL, improving the classification accuracy by 4.2% on average.

**Acknowledgment:** This work was supported in part by the Natural Science Foundation of China under Grant U23A20389, 62176139, and 62406177, in part by the Shandong Excellent Young Scientists Fund (Oversea) under Grant 2024HWYQ-027, in part by the Natural Science Foundation of Shandong province under Grant ZR2023QF124, in part by the Young Scholars Program of Shandong University, and in part by the Fundamental Research Funds of Shandong University.

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

# A    Appendix / supplemental material

## A.1    Training Algorithm

---

**Algorithm 1:** Training algorithm of the proposed VT-FSL.

---

**Input:** Training set $\mathcal{C}_{train}$, feature extractor $f_\theta$, linear projector $g_\eta$, two-layer perception $g_\varphi$

**while** *not converged* **do**

    1. Sample an $N$-way $K$-shot task $\mathcal{T} = \{\mathcal{S}, \mathcal{Q}\}$ from $\mathcal{C}_{train}$ by the episodic strategy;

    **for** *each class $y_i$ in $\mathcal{Q}$* **do**

        2. Generate textual descriptions $\mathcal{T}_N$ via LLM with class name and $K$ support images, using four structured reasoning stages: Strategy, Perception, Refinement, Conclusion;

        3. Generate $M$ synthetic images from $\mathcal{T}_N$ using a text-to-image model;

        4. Select top-$K$ images with LLM-based pairwise comparison according to Eq. 3;

    5. Extract support features $z_s$ and synthetic visual features $Z_v$ using $f_\phi$;

    6. Encode textual features $Z_t$ via CLIP, and project them to visual space with $g_\eta$;

    7. Fuse $Z_t$ and $z_s$ via $g_\varphi$ along channel and spatial dimensions, generating enhanced support features $Z_s$ according to Eq. 4, Eq. 5, and Eq. 6;

    8. Calculate the kernelized volume-based contrastive loss $\mathcal{L}_{\text{D2A}}$ and $\mathcal{L}_{\text{A2D}}$, jointly denoted as $\mathcal{L}_{\text{align}}$ with $Z_t$, $Z_s$, and $Z_v$ according to Eq. 10 and Eq. 11;

    9. Compute class prototypes $C = \{c_i\}_{i=1}^N$ by averaging $Z_s$ per class according to Eq. 1;

    **for** *each query image $x_j$ in $Q$* **do**

        10. Compute the prediction scores $p_{x_j}$ according to Eq. 2;

        11. Calculate the cross-entropy loss between $p_{x_j}$ and the ground-truth label $y_j$;

    12. Calculate the overall loss $\mathcal{L}_{\text{total}}$ according to Eq. 12;

    13. Update $\theta$, $\eta$, and $\varphi$ via gradient backpropagation

**Output:** Final prototypes with text and synthetic images for testing according to Eq. 13.

---

We describe the entire training procedure of VT-FSL in detail in Algorithm 1. The training process follows an episodic training paradigm and incorporates two key modules: (1) *Cross-modal Iterative Prompting (CIP)*, which generates class-specific textual descriptions based on both class names and support samples through a structured iterative reasoning process; and (2) *Cross-modal Geometric Alignment (CGA)*, which integrates textual, synthetic visual, and support features via a volume-based contrastive loss in a kernelized embedding space. These modules jointly enable the construction of cross-modal prototypes that are both semantically rich and visually grounded. The complete process, including prompt generation, feature extraction, fusion, prototype construction, loss computation, and parameter updates, is formalized in Algorithm 1.

## A.2    Text Generation Scheme

Some studies [24–28] extract additional semantic information from class names to compensate for the lack of representative semantics in limited support samples. However, class names typically provide minimal contextual information and are inherently ambiguous. For example, the term "mouse" may refer to either an animal or a computer device, offering little discriminative value without additional context.

To address this, SemFew [31] uses a large language model (LLM) to expand a class name into a detailed definition. However, to ensure textual-visual consistency, it relies on external knowledge bases such as WordNet [32], introducing additional matching constraints and reducing generation efficiency. ECER [30] also employs LLMs to generate a set of visual attributes from the class name, but due to the lack of guidance from support images, it requires a carefully designed attribute-filtering mechanism to remove visually irrelevant semantics.

In contrast, our method incurs no extra matching overhead and fully leverages the LLM's reasoning and generation capabilities to produce precise and visually grounded class descriptions. We propose a **Cross-modal Iterative Prompting (CIP)** framework, which conditions the LLM on both the class name and $K$ support images to elicit structured and grounded definitions. Specifically, CIP decomposes the generation process into four explicitly defined reasoning stages:

- **Strategy Stage**: Acts as a visual taxonomy expert, interpreting the class name to establish an initial semantic anchor and outlining the reasoning process.

- **Perception Stage**: Analyzes the $K$ support images to identify shared and discriminative visual traits that define the category.

- **Refinement Stage**: Refines the initial interpretation through logical reasoning, aligning class semantics with observed visual evidence, while eliminating hallucinated or irrelevant attributes.

- **Conclusion Stage**: Produces a concise, scientifically accurate, and generalizable definition, following a unified format and concrete example.

We guide the LLM to produce answers at each stage using a carefully designed prompt structure for iterative optimization of the final generated textual descriptions. The input is formatted using four special tags: `<STRATEGY></STRATEGY>`, `<PERCEPTION></PERCEPTION>`, `<REFINEMENT></REFINEMENT>`, and `<CONCLUSION></CONCLUSION>`, corresponding to outlining the problem, interpreting visual information, conducting reasoning, and producing the final definition, respectively.

The complete prompt template is shown below:

---

**Prompt Template for Cross-modal Iterative Prompting (CIP)**

**You are an expert in *computer vision* and *concept definition*.**
Given a class name and a set of representative images, your task is to generate a brief, scientifically accurate, and visually grounded definition. The definition should be primarily guided by the **semantic meaning of the class name**, and refined using the **visual evidence** from the images to resolve ambiguities and enhance relevance.
Please strictly follow the structured reasoning format below:
`<SUMMARY>`
Briefly describe your overall approach: (1) Interpret the class name and infer its possible meanings; (2) Analyze the image content to validate or refine the interpretation; (3) Formulate a revised description based on both sources. `</SUMMARY>`
`<CAPTION>`
Describe **common visual elements** observed across all images that help clarify the concept, such as objects, shapes, textures, colors, or background environments. `</CAPTION>`
`<REASONING>`
Explain how the visual patterns confirm or adjust the class meaning, especially if it is **ambiguous, polysemous, or abstract**. Use *step-by-step reasoning* to refine the concept. `</REASONING>`
`<CONCLUSION>`
Rewrite the class definition as a concise, scientifically sound, and visually consistent paragraph. Avoid redundancy; prioritize clarity and relevance.
*Example: The American robin is a widely recognized songbird, characterized by a rust-red to orange breast, dark grayish-blue upperparts, a white eye ring, and a slender yellow bill. It is commonly found in open woodlands and suburban gardens across North America.* `</CONCLUSION>`
**Class Name:** `{class_label}`
**Images:** `[K image inputs attached here]`

---

## A.3 Related Theories of Cross-modal Geometric Alignment

### A.3.1 Gram Matrix-based Volume

**Volume Computation Based on Gram Matrix.** Given a set of $k$ vectors $\mathbf{v}_1, \ldots, \mathbf{v}_k \in \mathbb{R}^n$, we aim to compute the volume of the $k$-dimensional parallelotope spanned by them. Let these vectors be arranged as columns in a matrix $\mathbf{A} = [\mathbf{v}_1, \ldots, \mathbf{v}_k] \in \mathbb{R}^{n \times k}$. The associated *Gram matrix* is defined as:

$$\mathbf{G}(\mathbf{v}_1, \ldots, \mathbf{v}_k) = \mathbf{A}^\top \mathbf{A} = \begin{bmatrix} \langle \mathbf{v}_1, \mathbf{v}_1 \rangle & \langle \mathbf{v}_1, \mathbf{v}_2 \rangle & \cdots & \langle \mathbf{v}_1, \mathbf{v}_k \rangle \\ \langle \mathbf{v}_2, \mathbf{v}_1 \rangle & \langle \mathbf{v}_2, \mathbf{v}_2 \rangle & \cdots & \langle \mathbf{v}_2, \mathbf{v}_k \rangle \\ \vdots & \vdots & \ddots & \vdots \\ \langle \mathbf{v}_k, \mathbf{v}_1 \rangle & \langle \mathbf{v}_k, \mathbf{v}_2 \rangle & \cdots & \langle \mathbf{v}_k, \mathbf{v}_k \rangle \end{bmatrix}. \tag{14}$$

This matrix captures all geometric information of the set $\{\mathbf{v}_1, \ldots, \mathbf{v}_k\}$, including vector lengths and pairwise angles. The $k$-volume of the parallelotope is computed by:

$$\mathrm{Vol}(\mathbf{v}_1, \ldots, \mathbf{v}_k) = \sqrt{\det \mathbf{G}(\mathbf{v}_1, \ldots, \mathbf{v}_k)}. \tag{15}$$

This formulation generalizes the intuition of Euclidean norm: when $k = 1$, the volume reduces to the norm of a single vector, i.e., $\mathrm{Vol}(\mathbf{v}) = \|\mathbf{v}\|$.

**Theoretical Validity.** To justify the above definition, we consider three cases:

- **Case I:** $k = n$. Then,

$$\det \mathbf{G}(\mathbf{v}_1, \ldots, \mathbf{v}_k) = \det(\mathbf{A}^\top \mathbf{A}) = (\det \mathbf{A})^2 = |\mathbf{A}|^2 \geq 0. \tag{16}$$

  Therefore,

$$\mathrm{Vol}(\mathbf{v}_1, \ldots, \mathbf{v}_k) = |\det \mathbf{A}| = \sqrt{\det \mathbf{A}^\top \mathbf{A}} = \sqrt{\det \mathbf{G}(\mathbf{v}_1, \ldots, \mathbf{v}_k)}. \tag{17}$$

- **Case II:** $k < n$. In this case, the volume lies within the $k$-dimensional subspace spanned by the vectors. Suppose $\{\mathbf{v}_1, \ldots, \mathbf{v}_k\}$ are linearly independent. We can extend this set to an orthonormal basis $\{\mathbf{w}_1, \ldots, \mathbf{w}_n\}$ of $\mathbb{R}^n$, and define an orthogonal transformation $\mathbf{O}$ that maps the basis to a new coordinate frame.

  The transformed vectors become:

$$\mathbf{v}_i' = \mathbf{O}(\mathbf{v}_i), \quad \mathbf{v}_i' = \begin{bmatrix} m_{1i} \\ m_{2i} \\ \vdots \\ m_{ki} \\ 0 \\ \vdots \\ 0 \end{bmatrix}, \tag{18}$$

  which visibly lie in a $k$-dimensional subspace of $\mathbb{R}^n$. Since orthogonal transformations preserve inner products, we have:

$$\langle \mathbf{v}_i', \mathbf{v}_j' \rangle = \langle \mathbf{v}_i, \mathbf{v}_j \rangle, \tag{19}$$

  and thus,

$$\mathrm{Vol}(\mathbf{v}_1, \ldots, \mathbf{v}_k) = \mathrm{Vol}(\mathbf{v}_1', \ldots, \mathbf{v}_k') = \sqrt{\det \mathbf{G}(\mathbf{v}_1, \ldots, \mathbf{v}_k)}. \tag{20}$$

- **Case III:** $k > n$. In this case, the vectors must be linearly dependent, and hence the $k$-dimensional volume is zero:

$$\det \mathbf{G}(\mathbf{v}_1, \ldots, \mathbf{v}_k) = 0, \quad \mathrm{Vol}(\mathbf{v}_1, \ldots, \mathbf{v}_k) = 0. \tag{21}$$

**Summary.** The determinant of the Gram matrix provides a robust and theoretically grounded measure of the volume of the parallelotope spanned by a set of vectors. This geometric quantity generalizes norm and angle in high-dimensional embedding spaces, and forms the basis of our proposed alignment loss for cross-modal representation matching.

### A.3.2 Volume as a Generalized Alignment Metric

We demonstrate that volume computation based on the Gram matrix provides a more comprehensive and expressive alignment mechanism compared to cosine similarity, particularly when multiple types of embeddings are involved.

We first consider the case of two vectors $\mathbf{v}_1, \mathbf{v}_2 \in \mathbb{R}^n$ with unit norm, i.e., $\|\mathbf{v}_1\| = \|\mathbf{v}_2\| = 1$. The Gram matrix is:

$$\mathbf{G} = \begin{bmatrix} \langle \mathbf{v}_1^\top, \mathbf{v}_1 \rangle & \langle \mathbf{v}_1^\top, \mathbf{v}_2 \rangle \\ \langle \mathbf{v}_2^\top, \mathbf{v}_1 \rangle & \langle \mathbf{v}_2^\top, \mathbf{v}_2 \rangle \end{bmatrix}, \tag{22}$$

with determinant:

$$\det(\mathbf{G}) = \langle \mathbf{v}_1^\top \mathbf{v}_1 \rangle \langle \mathbf{v}_2^\top \mathbf{v}_2 \rangle - \langle \mathbf{v}_1^\top \mathbf{v}_2 \rangle^2. \tag{23}$$

Hence, the volume spanned by these two vectors is:

$$\text{Vol} = \sqrt{\det(\mathbf{G})} = \sqrt{1 - \langle \mathbf{v}_1^\top \mathbf{v}_2 \rangle^2}. \tag{24}$$

Letting $\cos(\theta) = \langle \mathbf{v}_1, \mathbf{v}_2 \rangle$, this becomes:

$$\text{Vol} = \sqrt{1 - \cos^2(\theta)} = \sin(\theta). \tag{25}$$

This reveals that in the two-vector case, the volume is directly proportional to the sine of the angle between them, capturing the degree of geometric misalignment. Unlike cosine similarity, which increases as vectors become more aligned, volume achieves maximum when vectors are orthogonal, offering complementary information and sensitivity to structural differences.

**Volume Captures Full Pairwise Interactions.**   We extend the analysis to our setup involving three types of embeddings: textual features $\mathbf{T}$, support embeddings $\mathbf{S}$, and synthetic visual embeddings $\mathbf{V}$. The corresponding Gram matrix is:

$$\mathbf{G} = \begin{bmatrix} TT & TS & TV \\ ST & SS & SV \\ VT & VS & VV \end{bmatrix}, \tag{26}$$

where $TS = \langle \mathbf{T}, \mathbf{S} \rangle$, $TV = \langle \mathbf{T}, \mathbf{V} \rangle$, and so on. All embeddings are normalized such that $TT = SS = VV = 1$. The determinant of this matrix expands as:

$$\det(\mathbf{G}) = TT \cdot (SS \cdot VV - SV \cdot VS) - TS \cdot (ST \cdot VV - SV \cdot VT)$$
$$+ TV \cdot (ST \cdot VS - SS \cdot VT), \tag{27}$$

and simplifies to:

$$\det(\mathbf{G}) = 1 \cdot (1 - SV^2) - TS \cdot (TS - SV \cdot VT) + TV \cdot (TS \cdot SV - TV)$$
$$= 1 - SV^2 - TS^2 + TS \cdot SV \cdot VT + TV \cdot TS \cdot SV - TV^2$$
$$= 1 - SV^2 - TS^2 - TV^2 + 2 \cdot TS \cdot SV \cdot VT. \tag{28}$$

This result shows that volume-based computation naturally incorporates all pairwise interactions among the three embeddings—textual ($\mathbf{T}$), support ($\mathbf{S}$), and synthetic visual ($\mathbf{V}$). In particular, it accounts for relationships like $SV$, which are not explicitly considered when only using pairwise cosine similarities between the anchor ($\mathbf{T}$) and the other embeddings.

**Comparison with Cosine-based Alignment.**   In many existing methods, only the similarities between the anchor and the remaining embeddings (e.g., $TS$ and $TV$) are computed, while relationships among the non-anchor embeddings (e.g., $SV$) are ignored. This may lead to suboptimal alignment, especially when synthetic features or support embeddings deviate semantically.

In contrast, volume-based alignment considers all pairwise inner products in a unified geometric form. The determinant of the Gram matrix thus encodes higher-order consistency among the embeddings, enabling a more robust alignment objective that jointly optimizes their global compatibility.

### A.3.3 Kernelized Volume for Nonlinear Alignment

While the volume computed from the standard Gram matrix already captures higher-order geometric relationships among textual, support, and synthetic visual embeddings, it remains a linear measure. To further model complex nonlinear relationships, we extend our formulation to a high-dimensional Reproducing Kernel Hilbert Space (RKHS) via kernel embedding.

Let $\kappa : \mathbb{R}^n \times \mathbb{R}^n \to \mathbb{R}$ be a positive definite kernel function. The most common example is the Radial Basis Function (RBF) kernel, defined as:

$$\kappa(\mathbf{x}, \mathbf{z}) = \exp\left(-\frac{\|\mathbf{x} - \mathbf{z}\|^2}{2\sigma^2}\right), \tag{29}$$

where $\sigma$ controls the smoothness of the kernel. Such kernels define an implicit mapping $\phi : \mathbb{R}^n \to \mathcal{H}$ into a high-dimensional Hilbert space $\mathcal{H}$, where

$$\kappa(\mathbf{x}, \mathbf{z}) = \langle \phi(\mathbf{x}), \phi(\mathbf{z}) \rangle_{\mathcal{H}}. \tag{30}$$

Given a set of $k$ embeddings $\{\mathbf{v}_1, \dots, \mathbf{v}_k\}$, we construct a kernel Gram matrix $\mathbf{K} \in \mathbb{R}^{k \times k}$ as:

$$\mathbf{K}_{ij} = \kappa(\mathbf{v}_i, \mathbf{v}_j), \quad \forall i, j \in \{1, \dots, k\}. \tag{31}$$

Due to the positive definiteness of $\kappa$, the kernel Gram matrix $\mathbf{K}$ is symmetric and positive semi-definite. Thus, we can generalize the definition of volume in the RKHS as:

$$\mathrm{Vol}_{\mathcal{H}}(\mathbf{v}_1, \dots, \mathbf{v}_k) = \sqrt{\det(\mathbf{K})}. \tag{32}$$

This formulation retains the original motivation of measuring the mutual independence or spread among embeddings, but now in a nonlinear feature space where complex dependencies can be captured.

**Why Kernelized Volume Matters.** The kernelized volume metric $\mathrm{Vol}_{\mathcal{H}}$ has several key advantages over both cosine similarity and its linear volume counterpart:

- **Nonlinear feature interactions:** Through the kernel map $\phi$, we effectively transform the original embeddings into a high-dimensional space where nonlinear interactions become linearly separable. This enables detection of subtle structural mismatches between the three embeddings (text $\mathbf{T}$, support $\mathbf{S}$, synthetic vision $\mathbf{V}$) that may not be captured by inner product or cosine measures in the original space.

- **Higher expressive power:** Unlike cosine similarity, which only encodes directional alignment between pairs of vectors, the determinant $\det(\mathbf{K})$ captures how all $k$ embeddings interact geometrically in the kernel space. This includes all mutual pairwise kernel similarities and their higher-order arrangements.

- **Connection to independence and compactness:** When the embeddings $\phi(\mathbf{v}_1), \dots, \phi(\mathbf{v}_k)$ are highly correlated, the volume tends to zero, indicating collapse in representation diversity. Conversely, a larger volume suggests that the embeddings are geometrically well-distributed and carry complementary information. This is especially valuable for few-shot learning, where redundancy among features may severely harm generalization.

**Kernel Properties and Geometric Interpretation.** From the theory of positive definite kernels, we know that for any set of points $\{x_1, x_2, \dots, x_N\}$, the associated kernel matrix:

$$\mathbf{K} = [\kappa(x_i, x_j)]_{N \times N} \tag{33}$$

is guaranteed to be symmetric and positive semi-definite. This ensures that the square root of its determinant always yields a valid (possibly zero) volume in RKHS.

In essence, $\mathrm{Vol}_{\mathcal{H}}$ defines the volume of a parallelotope formed by $\phi(\mathbf{v}_1), \dots, \phi(\mathbf{v}_k)$ in $\mathcal{H}$:

$$\mathrm{Vol}_{\mathcal{H}}(\mathbf{v}_1, \dots, \mathbf{v}_k) = \|\phi(\mathbf{v}_1) \wedge \cdots \wedge \phi(\mathbf{v}_k)\|, \tag{34}$$

where $\wedge$ denotes the exterior product. This geometric interpretation highlights its capacity to quantify not just similarity, but high-dimensional structural diversity.

Table 8: The splits of categories and the number of categories/images in each few-shot dataset.

| Dataset | #(Class) | | | #(Image) | |
|---|---|---|---|---|---|
| | $\mathcal{D}_{\text{train}}$ | $\mathcal{D}_{\text{valid}}$ | $\mathcal{D}_{\text{test}}$ | Test | Total |
| *mini*ImageNet [12] | 64 | 16 | 20 | 12,000 | 60,000 |
| *tiered*ImageNet [60] | 351 | 97 | 160 | 206,209 | 779,165 |
| CIFAR-FS [67] | 64 | 16 | 20 | 12,000 | 60,000 |
| FC100 [19] | 60 | 20 | 20 | 12,000 | 60,000 |
| CUB-200-2011 [68] | 100 | 50 | 50 | 2,958 | 11,788 |
| Stanford-Cars [69] | 130 | 17 | 49 | 4,103 | 16,185 |
| Stanford-Dogs [70] | 70 | 20 | 30 | 5,115 | 20,580 |
| Places [71] | 183 | 91 | 91 | 18,200 | 73,000 |
| Plantae [72] | 100 | 50 | 50 | 17,253 | 68,461 |

**Summary.** By embedding the embeddings into a kernel-induced space and computing their geometric volume, our approach moves beyond simple pairwise similarity, capturing richer nonlinear relationships among text, support, and synthetic visual features. This kernelized extension of the CGA loss offers greater alignment flexibility and improves robustness under complex cross-space discrepancies.

## A.4 Experiments

### A.4.1 Dataset Details

We evaluate our method on a range of widely used few-shot learning benchmarks across three scenarios: standard few-shot classification, fine-grained few-shot classification, and cross-domain few-shot classification. Detailed statistics of all datasets are summarized in Table 8.

**(1) Standard Few-Shot Classification Benchmarks.** Following [23, 31, 47], we adopt the following four benchmark datasets:

- *mini*ImageNet [12]: A subset of ImageNet [95] consisting of 100 object categories, each with 600 images, totaling 60,000 images. The dataset is split into 64 classes for training, 16 for validation, and 20 for testing.

- *tiered*ImageNet [60]: A larger subset of ImageNet [95] comprising 779,165 images from 608 categories, grouped into broader semantic superclasses. The training/validation/test split contains 351, 97, and 160 categories, respectively.

- **CIFAR-FS** [67]: Derived from CIFAR-100 [96] by randomly splitting the 100 classes into 64 for training, 16 for validation, and 20 for testing. Each class contains 600 images, resulting in 60,000 images in total.

- **FC100** [19]: Also based on CIFAR-100 [96], but categories are split according to their semantic superclasses. It uses 60 classes from 12 superclasses for training, 20 classes from 4 superclasses for validation, and 20 classes from 4 different superclasses for testing. The large semantic gap across splits makes FC100 more challenging.

**(2) Fine-Grained Few-Shot Classification Benchmarks.** Following [80–82], we evaluate on three datasets focused on fine-grained visual categories:

- **CUB-200-2011 (CUB)** [68]: Comprises 11,788 images from 200 bird species. The classes are divided into 100 for training, 50 for validation, and 50 for testing. Each image is cropped to the human-annotated bounding box of the bird.

- **Stanford-Cars** [69]: Contains 16,185 images from classes of cars. The split includes 130 training classes, 17 validation classes, and 49 testing classes.

- **Stanford-Dogs** [70]: Consists of 20,580 images of 120 dog breeds. The dataset is split into 70 training, 20 validation, and 30 testing classes.

Table 9: The performance of different backbone architectures. *indicates our implementation.

| Method | Backbone | miniImageNet | | tieredImageNet | |
| --- | --- | --- | --- | --- | --- |
| | | 1-shot | 5-shot | 1-shot | 5-shot |
| ProtoNet [11] | Resnet-12 | $62.39_{\pm 0.21}$ | $80.53_{\pm 0.14}$ | $68.23_{\pm 0.23}$ | $84.03_{\pm 0.16}$ |
| ProtoNet* | Visformer-T | $62.48_{\pm 0.35}$ | $79.78_{\pm 0.26}$ | $68.85_{\pm 0.37}$ | $83.65_{\pm 0.28}$ |
| Meta-Baseline [22] | Resnet-12 | $63.17_{\pm 0.23}$ | $79.26_{\pm 0.17}$ | $68.62_{\pm 0.27}$ | $83.29_{\pm 0.18}$ |
| Meta-Baseline* | Visformer-T | $62.59_{\pm 0.34}$ | $79.88_{\pm 0.27}$ | $68.01_{\pm 0.35}$ | $82.75_{\pm 0.29}$ |
| ours | Visformer-T | $\mathbf{83.66_{\pm 0.31}}$ | $\mathbf{88.38_{\pm 0.25}}$ | $\mathbf{88.02_{\pm 0.34}}$ | $\mathbf{91.71_{\pm 0.27}}$ |

**(3) Cross-Domain Few-Shot Classification Benchmarks.** Following [84, 88, 91], we train the model on the *mini*ImageNet training set and evaluate it on domain-shifted datasets:

- **CUB** [68]: Same as above.

- **Places** [71]: contains 73,000 images with 365 scene categories. The classes are split into 183 base classes, 91 validation classes, and 91 test classes with 18,200 images.

- **Plantae** [72]: A subset of the iNaturalist [72] dataset focusing on plant species with 68,461 images. We follow the 100/50/50 split for training, validation, and testing classes, with 17,253 images in the test set.

### A.4.2 Ablation Study of Backbone Architectures

Table 9 showcases the performance comparison of different backbone architectures on the *mini*ImageNet and *tiered*ImageNet datasets under 1-shot and 5-shot settings. The results reveal that directly replacing the standard ResNet-12 backbone with the more advanced Visformer-T in existing baseline methods, such as ProtoNet and Meta-Baseline, fails to produce consistent performance gains. For example, in ProtoNet, the 1-shot accuracy only slightly improves from 62.39% to 62.48%, while the 5-shot accuracy even drops from 80.53% to 79.78%. A similar trend is observed for Meta-Baseline, where performance fluctuates around the original ResNet-12 values without notable improvement. This suggests that simply substituting the backbone does not effectively enhance the feature extraction or task-specific generalization ability of these methods.

In contrast, our method, VT-FSL, achieves a substantial performance boost when built upon the Visformer-T backbone. Specifically, VT-FSL attains 83.66% and 88.38% in 1-shot and 5-shot settings on *mini*ImageNet, and 88.02% and 91.71% on *tiered*ImageNet, outperforming all backbone-based baselines by a large margin. This significant improvement is attributed to the effective design of VT-FSL, which fully unleashes the potential of the backbone through cross-modal prompts and geometric-aware alignment. By leveraging rich cross-modal semantic information and global consistency among all features, VT-FSL dynamically enhances the extracted features and facilitates more discriminative class-specific representations, leading to superior generalization in few-shot scenarios.

### A.4.3 The Effect of Different Kernel Functions

Table 10: The effect of different kernel functions from kernelized volume-based contrastive loss

| Type | miniImageNet | | CIFAR-FS | | tieredImageNet | |
| --- | --- | --- | --- | --- | --- | --- |
| | 1-shot | 5-shot | 1-shot | 5-shot | 1-shot | 5-shot |
| Linear | $82.01_{\pm 0.30}$ | $87.10_{\pm 0.27}$ | $87.33_{\pm 0.33}$ | $90.25_{\pm 0.28}$ | $87.35_{\pm 0.36}$ | $90.07_{\pm 0.28}$ |
| Poly | $82.76_{\pm 0.31}$ | $87.80_{\pm 0.28}$ | $87.82_{\pm 0.33}$ | $89.87_{\pm 0.30}$ | $87.51_{\pm 0.36}$ | $90.70_{\pm 0.29}$ |
| RBF | $\mathbf{83.66_{\pm 0.31}}$ | $\mathbf{88.38_{\pm 0.25}}$ | $\mathbf{88.67_{\pm 0.32}}$ | $\mathbf{91.45_{\pm 0.28}}$ | $\mathbf{88.02_{\pm 0.34}}$ | $\mathbf{91.71_{\pm 0.27}}$ |

**The Effect of Kernel Functions in CGA**. We evaluate the impact of different kernel functions used in our kernelized volume-based contrastive loss. The three kernels considered are:

$$\kappa(\mathbf{x}, \mathbf{y}) = \begin{cases} \mathbf{x}^\top \mathbf{y}, & \text{Linear kernel} \\ (\mathbf{x}^\top \mathbf{y} + c)^d, & \text{Polynomial kernel (Poly)} \\ \exp\left(-\frac{\|\mathbf{x}-\mathbf{y}\|^2}{2\sigma^2}\right), & \text{Radial Basis Function (RBF)} \end{cases} \tag{35}$$

Table 10 reports the results on *mini*ImageNet, CIFAR-FS, and *tiered*ImageNet under both 1-shot and 5-shot settings. We observe that the RBF kernel consistently yields the best performance across all datasets. For instance, it achieves 83.66% and 88.38% on *mini*ImageNet, 88.67% and 91.45% on CIFAR-FS, outperforming the other two kernels.

The superior performance of the RBF kernel is attributed to its ability to project data into an infinite-dimensional Hilbert space, capturing fine-grained nonlinear relationships among textual, support, and synthetic visual embeddings. In contrast, the polynomial kernel introduces limited nonlinearity and is sensitive to hyperparameters such as degree and offset. The linear kernel performs the worst, as it lacks the expressiveness required to model complex cross-modal structures and effectively reduces to a dot product in the original feature space. These results validate the effectiveness of kernelized geometric alignment and confirm the choice of the RBF kernel as the default setting in our approach.

### A.4.4 The Effect of Anchor Selection in Contrastive Learning

Table 11: The effect of different Anchors from kernelized volume-based contrastive loss

| Type | *mini*ImageNet | | CIFAR-FS | | *tiered*ImageNet | |
|---|---|---|---|---|---|---|
| | 1-shot | 5-shot | 1-shot | 5-shot | 1-shot | 5-shot |
| Vision | $82.70_{\pm 0.30}$ | $87.60_{\pm 0.28}$ | $87.91_{\pm 0.35}$ | $90.58_{\pm 0.30}$ | $87.30_{\pm 0.36}$ | $90.92_{\pm 0.29}$ |
| Text | $\mathbf{83.66_{\pm 0.31}}$ | $\mathbf{88.38_{\pm 0.25}}$ | $\mathbf{88.67_{\pm 0.32}}$ | $\mathbf{91.45_{\pm 0.28}}$ | $\mathbf{88.02_{\pm 0.34}}$ | $\mathbf{91.71_{\pm 0.27}}$ |

**The Effect of Anchor Selection in Contrastive Learning.** We conduct an ablation study to investigate the effect of anchor selection in the kernelized volume-based contrastive loss. Specifically, we compare two anchor choices: (1) **Text**, where the textual feature $Z_t$ generated by the LLM serves as the anchor; and (2) **Vision**, where the synthetic visual feature $Z_v$, generated by the text-to-image model from the same class description, is used as the anchor. The results are reported in Table 11. As shown, using the textual feature as the anchor consistently leads to better performance across all datasets and settings. For instance, on *mini*ImageNet, the accuracy improves from 82.70% to 83.66% in the 1-shot setting and from 87.60% to 88.38% in the 5-shot setting when switching from the Vision to the Text anchor. Similar trends are observed on CIFAR-FS (88.67% vs. 87.91%) and *tiered*ImageNet (88.02% vs. 87.30%) under the 1-shot setting.

This result validates our default choice of **textual anchor** in contrastive learning. Although both generated text and synthetic images are used to enrich the support representation $Z_s$, the textual feature $Z_t$ provides a class-level semantic anchor that is more stable and discriminative. In contrast, the synthetic visual feature $Z_v$, while visually grounded, is more prone to noise and distributional artifacts due to limitations in generative models. Furthermore, aligning support and visual features around a textual anchor enables the model to centralize its representation around high-level semantics, which is particularly beneficial under few-shot scenarios where visual variability is high and labeled samples are scarce. Therefore, anchoring on the generated text helps better guide the cross-modal geometric alignment and leads to more robust class representations.

### A.4.5 The Effect of Different LLMs and Text-to-Image Models

We investigate the effect of large language models (LLMs) and Text-to-Image (T2I) Models on VT-FSL: LLM used for class description generation, and the text-to-image model used for synthetic image generation.

**Large language models.** As illustrated in Fig 9a, we compare GPT-4o [33] and Qwen2.5-VL-32B [34] under both 1-shot and 5-shot settings on *mini*ImageNet and CIFAR-FS. GPT-4o slightly outperforms Qwen2.5-VL-32B in 1-shot settings and maintains comparable or superior performance in 5-shot settings. The consistent results validate the robustness of VT-FSL across different LLMs,

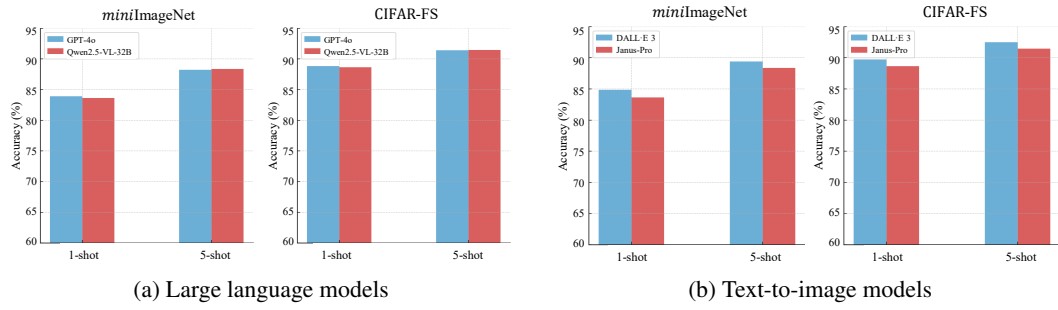

(a) Large language models          (b) Text-to-image models

Figure 9: The Effect of different LLMs and text-to-image models

but also suggest that GPT-4o's structured reasoning and richer world knowledge provide more semantically precise class descriptions, which are particularly beneficial when visual samples are scarce.

**Text-to-image models.** In Fig 9b, we examine the impact of two popular T2I models: DALL·E 3 [35] and Janus-Pro [36]. Results demonstrate that DALL·E 3 provides stronger visual priors than Janus-Pro, yielding higher accuracy across both datasets and settings. This is likely because DALL·E 3 generates more visually consistent and semantically faithful images from class-level descriptions, improving the quality of visual support features used during training. Despite Janus-Pro being multimodally capable, its generation quality is more variable and less optimized for fine-grained alignment with textual descriptions.

Overall, these results highlight that while VT-FSL is generally robust to backbone choices, the selection of higher-capacity and better-aligned LLMs and T2I models can further boost performance by improving the quality of generated semantic priors.

### A.5 Limitations

While VT-FSL demonstrates strong performance across a variety of few-shot classification benchmarks, several limitations remain. First, although we evaluate the method on cross-domain datasets such as Places and Plantae, these settings still exhibit certain similarities to the source domain. The robustness of VT-FSL under more challenging distribution shifts (e.g., medical images) has not been fully assessed. Second, VT-FSL depends on the quality of external generative models to provide textual descriptions and synthetic visual samples. While the proposed Cross-modal Iterative Prompting (CIP) fully leverages the reasoning and generation capabilities of large models to produce semantically rich descriptions, the overall quality of the generated content remains inherently bounded by the capacity of the underlying models. For example, weaker LLMs may produce generic or noisy descriptions, and low-quality image synthesis could introduce misleading visual signals. Finally, although we introduce a kernelized volume-based contrastive loss to enhance multimodal alignment, its theoretical behavior under high-dimensional, noisy, or semantically entangled feature distributions remains underexplored. Further study is needed to rigorously understand its convergence properties and sensitivity to kernel choice. We hope these observations inspire future improvements in robustness, interpretability, and generalization for few-shot multimodal learning.

