# OpenReview forum: "VT-FSL: Bridging Vision and Text with LLMs for Few-Shot Learning"
_NeurIPS.cc/2025/Conference — NeurIPS 2025 poster_

### Official Review · Reviewer_6u5o · 2025-06-26

**Clarity:** 3
**Significance:** 2
**Originality:** 2
**Rating:** 4
**Confidence:** 3

**Summary:**

This paper claims to be addressing the semantic hallucination problem when class names or LLM generated descriptions of classes are used in few-shot learning. The paper argues that features corresponding to LLM description of classes or the class attributes (generated based on the class names) must be aligned with the support images and generated synthetic images. The paper uses a prompting strategy to select the best K (e.g., 4) synthetic images from multiple images generated by JanusPro based on the textual description given by Qwn2.5-VL-32B using a pair-wise comparison. The vision encoder is Visformer-Tiny and the text encoder is ViT-B/16 CLIP. It aligns textual features, features of generated images, and the features of the support set images.

The results---standard few shot classification on miniImageNet and tiredImageNet, CIFAR-FS FC100, fine-grained few-shot classification on CUB-200-2022, Stanford-Dogs, Stanford-Cars, cross-domain few-shot classification on miniImageNet to CUB, miniImageNet to Places, miniImageNet to Plantae---are superior.

**Questions:**

1.	Related to items 1 and 2 above, justify the use of several large models. Report on the timing.
2.	Address the concern about the insufficient novelty.
3.	Improve clarity.

**Ethical Concerns:**

["NO or VERY MINOR ethics concerns only"]

**Final Justification:**

I have read through the comments by other reviewers, authors' rebuttals, and further responses. These discussions, in my opinion, substantially address the concerns. Therefore, I wish to retain my "Borderline accept" rating.

**Limitations:**

Reliance on the LLM and the image generator.

**Paper Formatting Concerns:**

Definitions of some parameters are missing (e.g., S_i) or given further away (e.g. z_s).
Section 3.3 is difficult to follow.

**Quality:**

3

**Strengths And Weaknesses:**

Strengths:

1.	Superior results.
2.	Prompting using the features of both support-set images and textual descriptions.
3.	Aligning the embeddings of textual descriptions, support image embeddings, and generated image embeddings.

Weaknesses:

1.	Using a large language model and a large image generator in addition to the Visformer-Tiny visual encoder gives an unfair advantage to this model. It would be important to compare training and inference times with a few existing papers.
2.	Picking the best K image generations probably gives very clean images representing the class in question. Fig. 6b shows such clean generations. The ablation in Table 5 shows that the contribution of the P_vision alone is the largest. This too is, in my opinion, an unfair advantage.
3.	What do the numbers in row 1 of Table 5 represent? Is it the few-shot classification accuracy of Visformer-Tiny?
4.	Using a carefully generated descriptive caption embedding with prototype embeddings exists in works cited in this paper, e.g., [27], and [26]. In this context, the novelties are 1. Cross-modal prompting, and 2. Geometric alignment. I am concerned whether this novelty is sufficient. Does the ablation show prompting with textual data only?

---

> ### Author Rebuttal · Authors · 2025-07-25
>
> We thank you for the constructive review and appreciation of our work such as “Superior results”. Below is our feedback. All concerns will be carefully addressed in the revised paper.
>
>
> > **Q: Using a large language model and a large image generator in addition to the Visformer-Tiny visual encoder gives an unfair advantage to this model. It would be important to compare training and inference times with a few existing papers.**
>
> Thank you for pointing out the concern about the potential unfair advantage of LLMs and generative models, suggesting a comparison of training and inference times.
> Several recent methods [20, 24, 26, 27] also leverage LLMs and generative models to generate textual descriptions and additional images to compensate for limited data. In contrast, the proposed VT-FSL utilizes the LLM to generate precise class descriptions iteratively in a single systematic and structured reasoning pass, enabling the zero-shot synthesis of semantically consistent images by the generative model.
> Moreover, the LLM and generative model are used only once in an offline prompt‑construction stage to construct cross‑modal prompts. After this stage, all downstream training/inference relies solely on the common lightweight Visformer‑T backbone (≈ 10 M parameters), without any dependence on large models. The cross‑modal prompts are integrated with support features as feature representations, adding virtually no extra overhead during training or inference.
>
> We compare the efficiency of training and inference times with a few existing FSL methods. This experiment is conducted on miniImageNet using the same server configuration (single NVIDIA RTX 6000 Ada 48G with two Intel 4410Y CPUs), compared to SP [19] without any large models to generate descriptions/images and SemFew [27] and ECER [26] with these models. SP and ECER employ the same Visformer-T backbone as VT-FSL during training/inference.
>
> | Method            | Prompt (h) | Training (min) | Inference (ms) | Acc (%) |
> | --- | --- | --- | - | -- |
> | SP \[19]          | --                      | 1.7                    | 78                     | 72.31                |
> | SemFew \[27]      | 1.5                     | 2.3                   | 105                     | 78.94               |
> | ECER \[26]        | 0.7                     | 3.0                    | 119                     | 81.14                |
> | **VT-FSL (ours)** | **0.7**                 | **1.1**                | **76**                 | **83.66**            |
>
> As shown in the table above, VT‑FSL achieves both the lowest training time (1.1 min) and inference time (76 ms), while attaining the highest accuracy (83.66%). Compared to ECER, VT‑FSL reduces training time by 63% (1.1 vs. 3.0 min) and inference time by 36% (76 vs. 119 ms), while improving accuracy by +2.5%. These results confirm that the plug-in design of VT-FSL transfers knowledge from large models efficiently in a one-time offline stage, and any downstream FSL model can directly use the resulting cross-modal prompts. This avoids the extra manual or algorithmic corrections required by SemFew and ECER, leading to a better trade‑off between efficiency and performance.
>
>
> > **Q: Picking the best K image generations probably gives very clean images representing the class in question. Fig. 6b shows such clean generations. The ablation in Table 5 shows that the contribution of the P_vision alone is the largest. This too is, in my opinion, an unfair advantage.**
>
> Thank you for pointing out the concerns about unfair advantage caused by visual prompts $P_{vision}$.
> Visual prompts $P_{vision}$ are generated based on the precise and visually grounded descriptions and filtered to top-K clean images with high high-quality by LLM-based pairwise comparison strategy.
> As shown in Table 5 of the manuscript, only $P_{vision}$ and only textual prompts $P_{text}$ consistently improves performance over the baseline by a large margin. However, when combing both $P_{vision}$ and $P_{text}$ significantly outperforms all ablations (e.g. from 79.17% of $P_{vision}$ to 82.08%), demonstrating the complementarity of textual and visual prompts.
>
> Moreover, combining table 5 of the manuscript, an additional ablation of w/o $P_{vision}$, top-K, and random-K selection is conducted. The results on CIFAR-FS and tieredImageNet are summarized as follows:
>
> | Method                               | CIFAR‑FS 1‑shot | CIFAR‑FS 5‑shot | tieredImageNet 1‑shot | tieredImageNet 5‑shot |
> | --- | -- | -- | -- | -- |
> | SemFew [27]| 84.34 ± 0.67 | 89.11 ± 0.54  |  82.37 ± 0.77 | 89.89 ± 0.52 |
> | ECER [26] | 86.01 ± 0.35 | --| 81.81 ± 0.51 | -- |
> | ours w/o $P_{vision}$ | 86.54 ± 0.34    | 89.74 ± 0.29    | 86.91 ± 0.37          | 90.19 ± 0.29          |
> | ours w/ random-K  $P_{vision}$   | 87.83 ± 0.34    | 91.01 ± 0.28    | 87.39 ± 0.34          | 91.09 ± 0.27          |
> | ours  w/ top-K  $P_{vision}$     | 88.67 ± 0.32    | 91.45 ± 0.28    | 88.02 ± 0.34          | 91.71 ± 0.27          |
>
> Even without $P_{\\text{vision}}$ (row 3), the proposed method already outperforms existing state-of-the-art approaches [26, 27]. Using $P_{\\text{vision}}$ with random-K selection still brings notable gains over these methods. These results and clarifications will be included into the revision.
>
>
> > **Q: What do the numbers in row 1 of Table 5 represent? Is it the few-shot classification accuracy of Visformer-Tiny?**
>
> | Setting            | $P_{\\text{text}}$ | $P_{\\text{vision}}$ | $\\mathcal{L}_{\\text{align}}$ | miniImageNet 1-shot | miniImageNet 5-shot | CIFAR-FS 1-shot | CIFAR-FS 5-shot |... |
> | -- | -- | -- | -- | - | -- | - | - | --- |
> | Baseline           | ✗       | ✗         | ✗        | 68.47 ± 0.43 | 82.63 ± 0.30 | 76.43 ± 0.45 | 86.60 ± 0.31 | ... |
>
> Thank you for pointing this out.
> The numbers in row 1 of Table 5 represent the few-shot classification accuracy of the baseline model, which uses the same Visformer-Tiny backbone but does not include any of our proposed modules: $P_{\\text{text}}$, $P_{\\text{vision}}$, and $\\mathcal{L}_{\\text{align}}$. This ensures a fair comparison, as all ablations and the full VT‑FSL model share identical backbone and training strategy. We will clarify this point in the revision.
>
>
> > **Q: Using a carefully generated descriptive caption embedding with prototype embeddings exists in works cited in this paper, e.g., [27], and [26]. In this context, the novelties are 1. Cross-modal prompting, and 2. Geometric alignment. I am concerned whether this novelty is sufficient. Does the ablation show prompting with textual data only?**
>
> Thank you for raising the concerns about the novelty of descriptive caption generation and the ablation of textual data only.
> First, as indicated in the introduction of the manuscript, several recent methods, SemFew [27] and ECER [26], generate descriptive captions based solely on class names, without leveraging the valuable visual patterns in the support images. This limitation can lead to semantic hallucinations and suboptimal descriptions, which requires costly manual or algorithmic corrections.
> Moreover, naive input prompting prevents LLMs from fully exploiting their reasoning capability.
> In contrast, the proposed VT-FSL method jointly leverages class names and support images via Cross-modal Iterative Prompting (CIP) to generate more visually grounded and precise descriptions through a single structured and systematic reasoning pass, followed by driving the synthesis of semantically consistent images.
>
> Regarding textual data only, the ablation studies in table 5 of manuscript clearly show that leveraging textual data alone already brings consistent performance gains over the baseline. A comparison of the results of textual data alone (ours w/o $P_{vision}$) with SemFew and ECER is shown below:
>
> | Method |CIFAR-FS 1-shot | CIFAR-FS 5-shot | tieredImageNet 1-shot | tieredImageNet 1-shot |
> |:----|:---|:---|:---|:--|
> | SemFew| $84.34 ± 0.67$ | $89.11 ± 0.54 $ | $ 82.37 ± 0.77$ | $89.89 ± 0.52$ |
> | ECER  | $86.01 ± 0.35$ | $ - $| $81.81 ± 0.51$ | $ - $ |
> | ours w/o $P_{vision}$ | $86.54 ± 0.34$ | $89.74 ± 0.29$ | $86.91 ± 0.37$ | $90.19 ± 0.29$ |
> | ours  | $\\bf{88.67} ± 0.34$ | $\\bf{91.45} ± 0.28$ | $\\bf{88.02} ± 0.34$ | $\\bf{91.71} ± 0.27$ |
>
> These superior results demonstrate that the Cross-modal Iterative Prompting (CIP) can provide richer and more precise textual semantics than existing methods [26, 27], thus still providing stronger discriminative capability even without synthetic images. Finally, when combined with complementary synthetic images and textual data, the proposed VT-FSL achieves best performance by providing high‑level class semantics and low‑level intra‑class diversity to compensate for limited support data.
>
>
> > **Q: Improve clarity. Definitions of some parameters are missing (e.g., S_i) or given further away (e.g. z_s). Section 3.3 is difficult to follow.**
>
> We would like to apologize if some parameters and the section 3.3 were not sufficiently clear.
> In the revised version, we will carefully clarify the parameters and improve the organization of the Section 3.3 to make it easier to understand.
> For example, $S_i$ is explicitly defined as the $i$‑th support category and support features $z_s$ are defined where they are first introduced to improve readability and avoid confusion.

---

> > ### Comment · Reviewer_6u5o · 2025-08-06
> > **Comments on Rebuttal**
> >
> > The authors have addressed my concerns about the use of several large models, the contribution of synthetic images, and fairness. I also noted that, as I can estimate, the authors have addressed the important comments of QnTs, cNcP.
> >
> > However, I think that the important concern of PMWQ, i.e., the image generation process compromising the integrity of the few-shot setting, has not been convincingly addressed.

---

> ### Author Response · Authors · 2025-08-06
> **Further details to address the concern that image generation may violate FSL integrity**
>
> Thank you sincerely for your constructive feedback. We are pleased that most of your concerns have been resolved with your approval.
>
> Regarding further concerns on the potential integrity problem of the few-shot setting caused by the image generation, we respectfully clarify that the image generation process of the proposed VT-FSL approach adheres strictly to the standard N-way K-shot protocol in few-shot learning (FSL). First, the N-way K-shot FSL protocol requires that models generalize to novel classes based on support samples per class, without access to additional real support samples. In VT-FSL, synthetic images are generated in a zero-shot fashion using only the class name and K support samples from each episode. No additional real data or unseen support information is accessed. Specifically, the Cross-modal Iterative Prompting (CIP) module generates class descriptions by conditioning on both class names and only K support samples from the support set $S_{N \times K}$, and the image generation process via text-to-image models operates in a zero-shot manner. This ensures that the enriched support set $\hat{S}_{N \times (K+K)}$ is derived from the same data budget and faithfully preserves the generalized low-data regime of few-shot learning. The K-shot generation number of VT-FSL is smaller than the methods [24,58] of the manuscript and the newly cited references [1–4].
>
> Secondly, to address sample scarcity, several prior works have explored image generation in few-shot learning, including CaFo [24] and ProtoDiff  [58] of the manuscript and some of the following. ProtoGAN [1] generates synthetic video features for novel classes by conditioning a CGAN on class prototypes. DISEF [2] leverages Stable Diffusion to synthesize class-consistent images from support samples. [3] explores the power of synthetic data from support examples to enhance few-shot object detection. [4] explores using synthetic samples to mitigate forgetting in few-shot class-incremental learning. These methods demonstrate that generating additional visual samples from limited support data is a community-approved and widely used strategy in few-shot learning research, which is fully consistent with the setting of the image generation process in the proposed VT-FSL. In contrast to previous FSL generation methods, our VT-FSL synthesizes semantically consistent images in a zero-shot manner by leveraging precise and visually grounded descriptions derived from the given support samples.
>
> References:
>
> [1] ProtoGAN: Towards Few Shot Learning for Action Recognition. In ICCV Workshop, 2019.
>
> [2] Diversified In-domain Synthesis with Efficient Fine-tuning for Few-Shot Classification. arXiv:2312.03046, 2023.
>
> [3] Explore the Power of Synthetic Data on Few-shot Object Detection. In CVPR, 2023.
>
> [4] Can Synthetic Images Conquer Forgetting? Beyond Unexplored Doubts in Few-Shot Class-Incremental Learning. arXiv:2507.13739, 2025.

---

### Official Review · Reviewer_PMWQ · 2025-06-29

**Clarity:** 3
**Significance:** 2
**Originality:** 3
**Rating:** 4
**Confidence:** 4

**Summary:**

This paper proposes a novel method named VT-FSL, which aims to bridge vision and text modalities using a large language model (LLM) to enhance few-shot learning (FSL). The authors argue that prototype-based FSL methods often suffer from non-discriminative class representations, resulting in semantic deviation from the true class centers. To address this, the proposed VT-FSL framework incorporates two key components: Cross-modal Iterative Prompting (CIP) and Cross-modal Geometric Alignment (CGA).

**Questions:**

All of my questions and concerns are outlined in the “Weaknesses” section above. If the authors are able to address these issues satisfactorily, I would be open to revising my score.

**Ethical Concerns:**

["NO or VERY MINOR ethics concerns only"]

**Final Justification:**

After reading the authors’ rebuttal, my concerns have been resolved. Therefore, I have raised my rating.

**Limitations:**

Yes

**Quality:**

3

**Strengths And Weaknesses:**

Strength:
1. The paper is well-structured and clearly motivated, making it easy to follow the authors' logic and contributions.
2. The idea of leveraging both visual and textual modalities for FSL is novel and timely, particularly with the growing capability of LLMs.
3. Experimental results demonstrate significant improvements over state-of-the-art methods, validating the effectiveness of the proposed approach.

Weakness:
1. The title of the paper is too generic and does not effectively convey the unique contributions or technical highlights of the work. A more specific and informative title would help better position the paper.
2. Some notations in the paper are unclear or inconsistent. For example,  Eq.(4) lacks the definition of the $z_v$. Additionally, the usage of $Z_t$ and $z_s$ and $Z_s$ is confusing—it's unclear why uppercase and lowercase letters are used differently. A clarification or unification of these notations is needed.
3. The paper would benefit from an ablation study on the loss function, e.g., evaluating the performance using only the cross-entropy loss, only $L_{D2A}$ and only $L_{A2D}$.
4. A major concern lies in the fairness of the experimental comparisons. The image generation process appears to introduce additional samples, which may compromise the integrity of the few-shot setting. Moreover, it is unclear whether the generative model has been exposed to the novel classes during training, which could further undermine the validity of the results. This point should be explicitly discussed and clarified.
5. The generation process seems to effectively double the number of shots in each class, which may lead to an unfair advantage. I strongly recommend the authors to conduct ablation studies that vary the number of generated images and report the impact on performance. This would help assess whether the performance gain is due to the proposed framework or simply the increased data volume.

---

> ### Author Rebuttal · Authors · 2025-07-25
>
> We thank your valuable comments and appreciation of our work such as “well-structured”, “clearly motivated”, “easy to follow”, and “novel and timely idea”. All concerns will be carefully addressed point by point in the revised paper. Below is our feedback:
>
>
> > **Q: The title of the paper is too generic and does not effectively convey the unique contributions or technical highlights of the work. A more specific and informative title would help better position the paper.**
>
> Thank you for pointing out the concern about the overly generic title.
> We agree that a more specific and informative title could further highlight our contributions. The current choice of title is deliberately chosen to balance informativeness and conciseness. The title "Bridging Vision and Text with LLMs" highlights the core idea of leveraging LLMs to construct precise cross‑modal prompts conditioned on both class names and support images and integrating them seamlessly.
> (1) The framework is designed as a plug‑in module: as shown in Appendix Fig. 8, the VT-FSL pipeline can be combined with other large models (e.g., GPT‑4o and DALL·E 3) to generate high‑quality cross‑modal prompts (e.g., 84.85% 1-shot accuracy on miniImageNet using DALL-E 3). Once constructed, these prompts can be directly utilized by any downstream training model in few-shot learning, providing strong flexibility and general applicability.
> (2) To the best of our knowledge, VT‑FSL is the first method to be extensively evaluated across three distinct FSL scenarios (standard, fine‑grained, and cross‑domain) on ten datasets, achieving an average improvement of 4.4%.
>
>
> > **Q: Some notations in the paper are unclear or inconsistent. For example, Eq.(4) lacks the definition of the $z_v$. Additionally, the usage of $Z_t$ and $z_s$ and $Z_s$ is confusing—it's unclear why uppercase and lowercase letters are used differently. A clarification or unification of these notations is needed.**
>
> Thank you for pointing out this issue. These notations will be carefully unified in the revision. For example, $z_v$ is corrected to $z_s$ in Eq. (4), as shown below:
> $\beta = \sigma \left( W_2 \, \sigma \left( W_1 \left[ Z_t ; \text{Avg}(z_s) \right] \right) \right),$
> where $Z_t$ denotes textual features encoded by the CLIP text encoder, the lowercase $z_s$ denotes the initial support features extracted by the visual backbone, and the uppercase $Z_s$ refers to the final enhanced support features after integrating textual features along spatial and channel dimensions.
> To further improve clarity, in the revised version, we will rename $z_s$ to $\bar{Z_s}$ and $Z_s$ to $\widetilde{Z_s}$ to clearly distinguish initial support features from the enhanced ones.
>
>
> > **Q: The paper would benefit from an ablation study on the loss function, e.g., evaluating the performance using only the cross-entropy loss, only $L_{D2A}$ and only $L_{A2D}$.**
>
> Thank you for the valuable suggestion.
> Only the cross-entropy (CE) loss is reported in table 5 (rows 1, 2, 4, and 6) of the manuscript, corresponding to baseline (CE), text + CE, vision + CE, and text + vision + CE.
>
> Additional ablation experiments are conducted using only $L_{D2A}$ and only $L_{A2D}$ on the miniImageNet and tieredImageNet datasets, as shown below:
>
> | $\\mathcal{L}_{\\text{D2A}}$ | $\\mathcal{L}_{\\text{A2D}}$  | miniImageNet 1-shot | miniImageNet 5-shot | tieredImageNet 1-shot | tieredImageNet 1-shot |
> |:-------|:---------|:----------------------|:----------------------|:------------------------|:------------------------|
> | ✗ | ✗  | $82.08 ± 0.31$ | $87.06 ± 0.27$ | $87.13 ± 0.36$ | $90.78 ± 0.28$ |
> | ✓ | ✗  | $83.16 ± 0.32$ | $88.01 ± 0.26$ | $89.32 ± 0.37$ | $91.31 ± 0.28$ |
> | ✗ | ✓ | $82.97 ± 0.23$ | $87.70 ± 0.17$ | $89.11 ± 0.27$ | $91.09 ± 0.18$ |
> | ✓ | ✓ | $83.66 ± 0.31$ | $88.38 ± 0.25$ | $88.02 ± 0.34$ | $91.71 ± 0.27$ |
>
> Textual, synthetic, and enhanced support features are transformed into normalized triplets $(Z_t, Z_v, Z_s)$.
> First, only CE the lowest accuracy, confirming that classification supervision cannot fully leverage multimodal information.
> Secondly, only $L_{D2A}$ significantly improves performance (e.g., 82.08% → 83.16% on miniImageNet 1-shot; 87.13% → 89.32% on tieredImageNet 1-shot), showing that text-anchored alignment effectively enhances global semantic consistency.
> Moreover, only $L_{A2D}$ also achieves clear gains but underperforms than $L_{D2A}$, suggesting that textual anchors provide stronger global semantic guidance.
> Finally, both $L_{D2A}$ and $L_{A2D}$ achieves the best results, confirming that bidirectional alignment offers complementary benefits and maximizes cross-modal feature integration. These new results and analyses will be included in the revised version.
>
>
> > **Q: A major concern lies in the fairness of the experimental comparisons. The image generation process appears to introduce additional samples, which may compromise the integrity of the few-shot setting. Moreover, it is unclear whether the generative model has been exposed to the novel classes during training, which could further undermine the validity of the results. This point should be explicitly discussed and clarified.**
>
> Thank you for pointing out the concerns about the potential fairness issue of additional samples and exposure of the generative model to novel classes.
> To alleviate the challenge of rare sample, several FSL methods [24, 58] have attempted to generate additional visual samples. Unlike relying on the class names in CaFo [24] or the support sample in ProtoDiff  [58] to generate, VT-FSL can generate samples with semantic consistency based on precise and visually grounded descriptions in a zero-shot manner. In addition, to ensure fairness, the VT-FSL generation process strictly follows the standard N-way K-shot FSL protocol, using only the given K-shot support samples and never touching any other real support samples.
>
> As shown in Fig. 5 of the manuscript, the effects of generated images using different textual templates are compared on miniImageNet, tieredImageNet, and CIFAR-FS. The average results are summarized below:
>
> | Text| 1-shot | 5-shot |
> |:-------|:----------------------|:----------------------|
> | ours (w/o $P_{vision}$)| $84.13$ | $88.76$ |
> | CaFo [24] | $81.91$ | $89.07$ |
> | SemFew [27] | $83.18$ | $89.20$ |
> | ours | $\\bf{86.78}$ | $\\bf{90.51}$ |
>
> Interestingly, using the template from CaFo [24] like “a photo of [CLASS]”  and class descriptions from SemFew [27] even leads to worse 1-shot performance than not using synthetic images (row 1) at all, confirming that the performance gains are not due to any prior exposure of the generative model to novel classes. Instead, the improvements come from precise cross‑modal prompts produced by CIP and geometry‑aware alignment (CGA), which fully exploit complementary textual and visual semantics.
>
>
> > **Q: The generation process seems to effectively double the number of shots in each class, which may lead to an unfair advantage. I strongly recommend the authors to conduct ablation studies that vary the number of generated images and report the impact on performance. This would help assess whether the performance gain is due to the proposed framework or simply the increased data volume.**
>
> Compared to the proposed VT-FSL, previous FSL works [24, 58] generate additional samples with more images or visual prototypes to enrich the limited support data. Moreover, as indicated in the above reply, the image generation process relies only on the given support images and never touches any additional real images. The entire process strictly follows the standard N-way K-shot setup in FSL [19-27], thereby ensuring fairness.
>
> As shown in Fig. 6a of the manuscript, the ablation study of different generated images per class was conducted on miniImageNet, tiereredImageNet, and CIFAR-FS. The results are summarized below. It can be observed that adding top 1–3 generated images leads to a noticeable performance gain, while adding more images causes the accuracy to plateau or even slightly decline. This trend is consistent across datasets. Excessive generated samples introduce noise, as lower-ranked generations are less discriminative and can degrade representation quality. This validates the effective design of selecting top-K generated images for K-shot tasks, which augments the support set without compromising the few-shot regime. Consequently, these results demonstrate that the performance gain comes from the framework design rather than simply increasing data volume.
>
> | Generated number            |  miniImageNet | tiereredImageNet  | CIFAR-FS |
> | ------------------ | ------- | --------- | -------- |
> | 0          | 78.96       | 86.91        | 86.54        |
> | 1              |  83.66      | 88.02        | 88.67  |
> | 2        | 84.01       | 88.07        | 88.93        |
> | 3            | 84.20      | 88.16        | 88.91     |
> | 4        |83.98 | 88.04         | 89.21     |
> | 5       | 84.0       | 88.02         | 89.01      |
> | 10 | 83.06       | 87.52        | 88.51     |

---

> > ### Comment · Reviewer_PMWQ · 2025-08-04
> >
> > I appreciate the authors’ comprehensive rebuttal. The additional justifications and ablation studies, especially those addressing fairness concerns, have adequately resolved the issues I previously raised. For the benefit of future readers, I encourage the authors to integrate these clarifications into the main manuscript. In light of these revisions, I am inclined to raise my overall evaluation score.

---

> ### Author Response · Authors · 2025-08-04
> **Thanks for positive response**
>
> Thank you very much for your positive response and for taking the time to review our rebuttal! We are pleased to hear that our rebuttal and the additional experiments have addressed your concerns. We sincerely appreciate your recognition of our work and the improved evaluation score. We will revise our paper in the final version.

---

### Official Review · Reviewer_cNcP · 2025-07-02

**Clarity:** 3
**Significance:** 2
**Originality:** 3
**Rating:** 4
**Confidence:** 4

**Summary:**

This paper uses a large language model and a text-to-image generation model to produce precise category descriptions and visual cues, serving as auxiliary information to enhance the support set images. Specifically, two key modules: Cross-modal Iterative Prompting, which utilizes category names and images to generate detailed textual descriptions and synthesized images; and Cross-modal Geometric Alignment, which employs a kernel-based volume minimization approach to align the representations across modalities are introduced. Experimental results validate the effectiveness of the proposed approach.

**Questions:**

Please refer to the weaknesses.

**Ethical Concerns:**

["NO or VERY MINOR ethics concerns only"]

**Final Justification:**

Thank you for the authors' detailed rebuttal, which resolved most of my technical concerns. Therefore, I maintain my original positive score.

**Limitations:**

Yes.

**Paper Formatting Concerns:**

No.

**Quality:**

3

**Strengths And Weaknesses:**

**Strengths**:
1. While previous works have primarily focused on incorporating auxiliary semantic information, this paper introduces the novel perspective of additionally leveraging visual information.
2. To better align support images, descriptive texts, and synthesized images, the paper proposes an interesting volume-based alignment method, which offers mathematical interpretability.
3. The experimental evaluation is relatively comprehensive and complete, demonstrating strong performance across benchmarks.

**Weaknesses**:
1. Regarding the selection of top-k synthesized images, how is the reliability of the selection ensured? The paper employs a LLM as a discriminator to perform pairwise comparisons. However, I wonder whether this process involves the support set at all. If not, can the visual consistency between the generated images and the support set images truly be guaranteed?
2. I understand this method as an attempt to transfer knowledge from large models to smaller ones, thereby enriching existing representations. The integration of a large model undoubtedly introduces additional computational overhead during training. I am curious about how the paper addresses or justifies this concern.
3. The term fusion factor k can be easily confused with k-shot. Moreover, the paper lacks an ablation study on this hyperparameter. I would like to know how the model performance changes as k varies from 0 to 1. Additionally, what is the optimal value of k for each dataset? These details could be included in the supplementary material.
4. Since your model internally utilizes the CLIP architecture, why not directly perform parameter-efficient fine-tuning based on CLIP?

---

> ### Author Rebuttal · Authors · 2025-07-25
>
> We sincerely thank you for your thoughtful review and positive recognition of our work, such as “novel perspective”, “interesting volume-based alignment method”, and “strong performance”. Below, we provide a point-by-point response to each of your comments:
>
>
> > **Q:  Regarding the selection of top-k synthesized images, how is the reliability of the selection ensured? The paper employs a LLM as a discriminator to perform pairwise comparisons. However, I wonder whether this process involves the support set at all. If not, can the visual consistency between the generated images and the support set images truly be guaranteed?**
>
> Thank you for raising the concern about ensuring the reliability of the synthesized images and their visual consistency with the support images. For each category, synthetic images are generated in a zero‑shot manner by feeding the textual description into a text‑to‑image generative model. The description, derived from both class names and support images, provides precise and visually grounded semantics, thereby facilitating the generation of images that share visual patterns with the support images. In addition, we introduce an LLM‑based pairwise comparison method that compares synthesized images in pairs to assess how well they match the textual description in terms of visual features. The images are then ranked accordingly, and the top‑K ones are selected to further enhance the reliability of generation.
>
> To validate the effectiveness of top-K selection, as shown in Fig. 6a of the manuscript, it can be observed that adding the top-ranked 1-3 synthesized images consistently improves average accuracy to 3.3%, while adding lower‑ranked ones reduces by 0.9%.
> To visualize visual consistency, Fig. 6b shows that the synthesized image is very consistent with the corresponding support image on the target object and carries less background noise.
> To further validate, an additional analysis is conducted on miniImageNet, where $S_i$ and $G_i$ respectively denote support and generated images of the i-th category. For each category, the cosine similarities between the generated image and the support image for each category are computed, as shown below:
>
> |Similarity| $S_1$ | $S_2$ | $S_3$ | $S_4$ |
> |:-------|:----------------------|:----------------------|:----------------------|:----------------------|
> | $G_1$ | **0.87** | $0.12$ |  $0.01$ |  $0.00$ |
> | $G_2$ | $0.01$ | **0.74** | $0.25$ |  $0.00$ |
> | $G_3$ | $0.02$ | $0.02$ | **0.65** |  $0.31$ |
> | $G_4$ | $0.01$ | $0.00$ | $0.29$ |  **0.70** |
>
> After applying a temperature‑scaled softmax to each row, the resulting 4 × 4 similarity matrix shows that the generated image of each category achieves its highest similarity with the corresponding support image, confirming strong visual consistency and reliability of generated images. For example, $G_1$ from the first category aligns best with $S_1$ at  0.87, while other similarities are minimal. This additional analysis will be added to the final revision.
>
>
> > **Q:  I understand this method as an attempt to transfer knowledge from large models to smaller ones, thereby enriching existing representations. The integration of a large model undoubtedly introduces additional computational overhead during training. I am curious about how the paper addresses or justifies this concern.**
>
> Thank you for raising the concern about the additional computational overhead from LLMs during training. In the proposed VT-FSL, the large model is used only once in an offline prompt‑construction stage to construct cross‑modal prompts. After this stage, all downstream training/inference relies solely on the common lightweight Visformer‑T backbone (≈ 10 M parameters), without any dependence on large models. The cross‑modal prompts are integrated with support features as feature representations, adding virtually no extra overhead during training or inference.
>
> To further validate the efficiency, additional experiments of computational overhead are conducted on miniImageNet using the same server configuration (single NVIDIA RTX 6000 Ada 48G with two Intel 4410Y CPUs), compared to SP [19] without large models to generate textual descriptions and SemFew [27] and ECER [26] with large models. SP and ECER employ the same Visformer-T backbone as VT-FSL during training.
>
> As shown in the table below, VT‑FSL achieves the lowest training (1.1 min) and inference time (76 ms) while obtaining the highest accuracy (83.66%). In particular, compared to ECER [26], VT‑FSL reduces training time by 63% (1.1 vs. 3.0 min) and inference time by 36% (76 vs. 119 ms), while improving accuracy by +2.5%. This improvement is attributed to the plug-in nature of VT-FSL, where cross-modal prompts are constructed once offline and can be directly used by any downstream training FSL model without training or additional computation, avoiding the additional manual or algorithmic corrections of SemFew [27] and ECER [26].
>
> | Method            | Prompt (h) | Training (min) | Inference (ms) | Acc (%) |
> | ----------------- | ----------------------- | ---------------------- | ---------------------- | ------------------ |
> | SP \[19]          | --                      | 1.7                    | 78                     | 72.31                |
> | SemFew \[27]      | 1.5                     | 2.3                   | 105                     | 78.94               |
> | ECER \[26]        | 0.7                     | 3.0                    | 119                     | 81.14                |
> | **VT-FSL (ours)** | **0.7**                 | **1.1**                | **76**                 | **83.66**            |
>
>
>
> > **Q: The term fusion factor k can be easily confused with k-shot. Moreover, the paper lacks an ablation study on this hyperparameter. I would like to know how the model's performance changes as k varies from 0 to 1. Additionally, what is the optimal value of k for each dataset? These details could be included in the supplementary material.**
>
> Thank you for pointing out the potential confusion caused by the notation of the fusion factor k and suggesting an ablation study for it.
> First, to avoid confusion with k-shot, the fusion factor will be renamed from $k$ to $u$ in the final version.
> Secondly, the fusion factor  $u$ controls the relative weight between textual and visual features to obtain the final inference prototypes in Eq. (13) of the manuscript. For each dataset, $u$ is automatically determined by a grid search over [0, 1] on the corresponding validation set, selecting the value that yields the highest validation accuracy. The optimal values are 0.5 (miniImageNet), 0.7 (tieredImageNet), 0.6 (CIFAR‑FS), and 0.6 (FC100).
>
> As shown in the table below, a full ablation study is conducted by varying $u$ from 0 to 1 with a step size of  0.1 in the 5‑way 5‑shot setting. The results show that performance peaks at intermediate values of $u$ , confirming that textual and visual features are complementary. Using only visual prompts ($u$= 0) or only textual prompts ($u$ = 1) consistently leads to inferior results. These ablation results, together with the optimal $u$ values, will be included in the supplementary material of the final version.
>
> | value of u| miniImageNet | tieredImageNet | CIFAR-FS | FC100 |
> |:-------|:---------------|:--------|:---------|:---------|
> | 0 | 86.92 | 89.68 |  90.43 |  65.07 |
> | 0.1 | 87.36| 90.22  |  90.68 |  65.82 |
> | 0.2 | 87.69 | 90.67 |  90.93 |  66.41 |
> | 0.3 | 88.03 | 91.06 |  91.14 |  66.96 |
> | 0.4 | 88.26 | 91.37 |  91.31 |  67.39 |
> | 0.5 | **88.38** | 91.57 |  91.41 | 67.57 |
> | 0.6 | 88.38 | 91.68 |  **91.45** |  **67.68** |
> | 0.7 | 88.32 | **91.71** |  91.40 |  67.49 |
> | 0.8 | 88.16 | 91.58 |  91.25 |  67.06 |
> | 0.9 | 87.91| 91.34 |  91.00 |  66.40 |
> | 1.0 | 87.55 | 90.96 |  90.63 |  65.54 |
>
>
> > **Q: Since your model internally utilizes the CLIP architecture, why not directly perform parameter-efficient fine-tuning based on CLIP?**
>
> Thank you for pointing out this insightful question.
> First, following the same settings [20, 24, 26, 27], the proposed VT-FSL only uses the frozen CLIP text encoder to extract textual embeddings, ensuring a fair comparison with prior FSL methods that do not fine‑tune CLIP.
> Secondly, conventional FSL approaches [19-27] assume the training pipeline consists of pre‑training and meta‑tuning. Directly fine‑tuning CLIP breaks this setting, as CLIP is pre‑trained on hundreds of millions of image‑text pairs. It becomes difficult to disentangle whether gains come from algorithmic design or access to massive external pre-training data, leading to potential unfair comparisons.
> Moreover, CLIP models such as ViT‑B/16 (≈ 86 M parameters) are ~8× larger than typical FSL backbones (e.g., Visformer-T ≈ 10 M), significantly increasing computational cost.
>
> As shown in the table below, we replace Visformer‑T with CLIP ViT‑B/16 and perform both LoRA fine‑tuning and full fine‑tuning. While LoRA fine‑tuning yields only a marginal gain (+0.4% accuracy), it requires ~10× longer training time (10.4 min vs. 1.1 min) and ~5× more GPU memory (22.1 GB vs. 4.3 GB). Full fine‑tuning incurs even higher costs (14.8 min, 28.6 GB). These results confirm that parameter‑efficient fine‑tuning of CLIP is computationally expensive and less suitable under the conventional few-shot benchmarks, making it difficult for deployment.
>
> | Backbone     | Params (M) | Fine-tuning Strategy | Acc (%) | Training (min) | GPU Memory (GB) |
> |-----------------------------|------------|----------------------|--------------|-----------------------|-----------------|
> | Visformer-T| 10.8       | Meta-training | 83.66    | **1.1**           | **4.3**         |
> | CLIP ViT-B/16| 86.7   | Adapter (LoRA)       | 84.02        | 10.4              | 22.1            |
> | CLIP ViT-B/16   | 86.7       | Full Fine-tuning     | 84.45        | 14.8              | 28.6            |

---

> > ### Comment · Reviewer_cNcP · 2025-08-05
> > **Official Comment by Reviewer cNcP**
> >
> > Thank you for the detailed rebuttal, which resolved most of my technical concerns. I will maintain my original score, as it reflects my overall assessment of the paper's contribution.

---

> ### Author Response · Authors · 2025-08-05
> **Thanks for feedback**
>
> Thank you very much for your valuable reviews and for acknowledging that our rebuttal resolved most of your technical concerns. We sincerely appreciate the time and effort you invested in reviewing our paper. We will revise our paper in the final version.

---

### Official Review · Reviewer_QnTs · 2025-07-04

**Clarity:** 3
**Significance:** 3
**Originality:** 2
**Rating:** 4
**Confidence:** 3

**Summary:**

The paper provides a novel way to efficiently extract semantic features from an image in a few-shot regime. Authors perform this extraction by proposing an approach containing two novel procedures: first, they augmentate the available examples by providing an iterative procedure to extract the semantic features of an image as text and use them to generate a new semantic-grounded image; second, they provide a geometry-based loss function to extract those features in a way they are aligned with the semantics of the class.

**Questions:**

* I do not understand table 5. Does each row represent a different baseline with the specified properties (i.e., text/vision/aling)? In that case, an additional column with the name would be handy. If not, does each row correspond to the same pre-training stage and different fine-tuning procedures? In this case, that may help to clarify the observation made in the "weaknesses" section.

* Baselines used for each table differ in some cases. Is there any particular reason for this? For instance, SemFew and ECER are the second best methods in table 2 but they don't appear in table 3. Similarely, SUITED is the second best method of table 3 but it is not shown in table 4. Is there any reason why those methods could not be used in other scenarios?

**Ethical Concerns:**

["NO or VERY MINOR ethics concerns only"]

**Final Justification:**

I thank the authors for their thoughtful and detailed rebuttal. The clarifications on Section 3.3 and Figure 2 will indeed improve readability, and the additional details on parameter sharing and the role of β are welcome. I also appreciate the careful explanation regarding backbone fairness and the inclusion of ablation studies, which strengthen the claims of the paper. Similarly, the planned improvements to Table 5 and the justification for the selection of baselines address several of my concerns about clarity and fairness.
That said, while the rebuttal provides convincing clarifications, my original concerns regarding the comparability of baselines and the interpretability of some results are only partially alleviated. I remain positive about the contributions and overall quality of the work, I have updated my score to reflect this improved understanding and appreciation of the paper’s contributions.

**Limitations:**

Yes, the authors adequately addressed the limitations and potential negative societal impact of their work.

**Paper Formatting Concerns:**

There is no major formatting issues in this paper

**Quality:**

3

**Strengths And Weaknesses:**

= Strengths:
* The approach provided by the authors is (as to my knowledge) novel and well implemented, although it can be seen as a more complex version of the work proposed in https://arxiv.org/pdf/2311.18649 .
* The overall framework (backbone + pre-training + fine-tuning) is unarguably better than the baselines shown.

= Weaknesses:
* I found it a little difficult to understand the overall system and I think that its description (mainly section 3.3 "Cross modal fusion") can be improved. For instance, it is not clear from figure 2 if the "Block" is shared in all cases, or where the $\beta$ vector appears on the figure. Other than that, the paper is generally well written.
* My main concern is the conclusions taken by comparing with the used baselines. I understand that they are taken directly from the corresponding table of the previous works, but there seems to be a lot of variance in the pre-training data, the backbone model and the fine-tuning procedure. That makes it difficult to determine if the framework is better because, for instance, the tieredImageNet was used in the pre-training stage or because of the loss function designed. As such, claims like the ones in line 213-214 ("These results highlight the effectiveness and the advantage of constructing complementary multimodal prompts with global alignment across all features for full semantic integration.") are not necessarily true, since there is (as to my understanding) no direct comparison between different fine-tuning procedures for the same pre-training stage.

---

> ### Author Rebuttal · Authors · 2025-07-25
>
> We sincerely appreciate your valuable reviews and positive recognition of our work (e.g., "novel and well-implemented approach", "generally well-written"). All concerns will be carefully addressed point by point in the revised paper. Below is our feedback:
>
> > **Q: I found it a little difficult to understand the overall system and I think that its description (mainly section 3.3 "Cross modal fusion") can be improved. For instance, it is not clear from figure 2 if the "Block" is shared in all cases, or where the β vector appears on the figure. Other than that, the paper is generally well written.**
>
> We would like to apologize if the description (mainly Section 3.3 "Cross modal fusion") has not been explicit enough and might thus have led to increased difficulty in understanding our work.
> In the revision, this section will be carefully revised to clearly indicate the parameter sharing of the "Block" and the β vector in Fig. 2 to improve readability.
> Specifically, first, as indicated in the caption of Fig. 2, the multiple Transformer blocks annotated with shared weights are served as the visual feature extractor. Moreover, we will clearly show in Fig. 2 where the β vector is computed and applied, as defined in Eq. (4):
>
> $\beta = \sigma \left( W_2 \, \sigma \left( W_1 \left[ Z_t ; \text{Avg}(z_s) \right] \right) \right),$
>
> where β is computed by integrating the textual features $Z_t$ and the average support features  $Avg(z_s)$ via the two-layer MLP. The resulting β dynamically modulates support features $z_s$ along the channel dimension via token‑wise scaling.
>
>
> > **Q: My main concern is the conclusions taken by comparing with the used baselines. I understand that they are taken directly from the corresponding table of the previous works, but there seems to be a lot of variance in the pre-training data, the backbone model and the fine-tuning procedure. That makes it difficult to determine if the framework is better because, for instance, the tieredImageNet was used in the pre-training stage or because of the loss function designed. As such, claims like the ones in line 213-214 ("These results highlight the effectiveness and the advantage of constructing complementary multimodal prompts with global alignment across all features for full semantic integration.") are not necessarily true, since there is (as to my understanding) no direct comparison between different fine-tuning procedures for the same pre-training stage.**
>
> Thanks for raising this concern about the fair comparisons in terms of pre‑training data, backbone, and fine‑tuning procedure.
> First, the proposed VT-FSL method adopts the same pre-training data and training procedure as in previous works [7–27].
> Specifically, VT-FSL follows the standard two-stage training framework, consisting of pre-training and fine-tuning on the training set of datasets (e.g., tieredImageNet) and is evaluated on the testing set, where both sets are disjoint.
> Secondly, VT‑FSL employs the widely used backbone Visformer‑T (≈ 10.0 M parameters), consistent with several previous works including SUN [13], SP [19], KTPP [20], and ECER [26]. The lightweight Visformer-T in terms of parameters is smaller than ResNet‑12 (≈ 12.4 M) and even several times smaller than other backbones (i.e., ViT-S/16 ≈ 22.0  M, Swin-T ≈ 29.0 M, and WRN-28-10 ≈ 36.5 M) to avoid unfair advantage from model capacity.
>
> Finally, an ablation study with different backbone architectures is conducted in Table 8 of the Appendix.  The results are reported on miniImageNet and tieredImageNet under 1-shot and 5-shot settings, summarized below:
>
> | Method | Backbone | miniImageNet 1-shot | miniImageNet 5-shot | tieredImageNet 1-shot | tieredImageNet 1-shot |
> |:-------|:---------|:----------------------|:----------------------|:------------------------|:------------------------|
> | ProtoNet| Resnet-12 | $62.39 ± 0.21$ | $80.53 ± 0.14$ | $68.23 ± 0.23$ | $84.03 ± 0.16$ |
> | ProtoNet * | Visformer-T | $62.48 ± 0.35$ | $79.78 ± 0.26$ | $68.85 ± 0.37$ | $83.65 ± 0.28$ |
> | Meta-Baseline| Resnet-12 | $63.17 ± 0.23$ | $79.26 ± 0.17$ | $68.62 ± 0.27$ | $83.29 ± 0.18$ |
> | Meta-Baseline * | Visformer-T | $62.59 ± 0.34$ | $79.88 ± 0.27$ | $68.01 ± 0.35$ | $82.75 ± 0.29$ |
> | ours | Visformer-T | $\\bf{83.66} ± 0.31$ | $\\bf{88.38} ± 0.25$ | $\\bf{88.02} ± 0.34$ | $\\bf{91.71} ± 0.27$ |
>
> $*$ denotes results produced by Visformer‑T. It can be clearly observed that merely replacing ResNet-12 with Visformer-T fails to improve results. Instead, VT-FSL significantly improves accuracy by an average of 20.0% and 7.8% under 1-shot and 5-shot settings, respectively.  This demonstrates that the proposed modules can fully leverage the unique advantages of ViT-based Visformer-T in long-range dependencies and cross-modal interactions by cross-modal prompts and geometric-aware alignment. Therefore, the claim in lines 213–214 is grounded on the fair experimental settings and is confirmed by superior performance of VT-FSL. More analysis is detailed in the Appendix.
>
>
> > **Q: I do not understand table 5. Does each row represent a different baseline with the specified properties (i.e., text/vision/aling)? In that case, an additional column with the name would be handy. If not, does each row correspond to the same pre-training stage and different fine-tuning procedures? In this case, that may help to clarify the observation made in the "weaknesses" section.**
>
> Thank you for raising the concern about the meaning of each row in Table 5 and suggesting an additional column with name to improve clarity.
> Each row represents an ablated variant of VT-FSL, where the textual prompts $P_\text{text}$, visual prompts $P_\text{vision}$, and kernelized volume-based contrastive loss $\mathcal{L}_\text{align}$ are selectively enabled or disabled.
> For example, row 1 of Table 5 denotes the baseline without any of the proposed modules of VT-FSL.
> It is worth noting that results of all variants are built on the same backbone and training strategy, ensuring that the comparison isolates the contribution of each module.
>
> In the final revision, an additional "Setting" column will be included to explicitly name each variant, as shown below:
>
> | Setting            | $P_{\\text{text}}$ | $P_{\\text{vision}}$ | $\\mathcal{L}_{\\text{align}}$ | ... |
> | ------------------ | ------- | --------- | -------- | --- |
> | Baseline           | ✗       | ✗         | ✗        | ... |
> | +Text              | ✓       | ✗         | ✗        | ... |
> | +Text+Align        | ✓       | ✗         | ✓        | ... |
> | +Vision            | ✗       | ✓         | ✗        | ... |
> | +Vision+Align        |✗        | ✓         | ✓        | ... |
> | +Text+Vision       | ✓       | ✓         | ✗        | ... |
> | +Text+Vision+Align | ✓       | ✓         | ✓        | ... |
>
>
> > **Q: Baselines used for each table differ in some cases. Is there any particular reason for this? For instance, SemFew and ECER are the second best methods in table 2 but they don't appear in table 3. Similarely, SUITED is the second best method of table 3 but it is not shown in table 4. Is there any reason why those methods could not be used in other scenarios?**
>
> Thank you for pointing the concern about why many strong methods are not present in all SOTA experiments. The results of the existing methods in all SOTA tables are taken directly from their original papers to ensure a fair comparison.  However, strong methods such as SemFew, ECER, and SUITED are not evaluated across all scenarios in their papers.For example, SemFew and ECER have not been evaluated on fine‑grained benchmarks, whereas SUITED does not report results on cross‑domain scenarios. In contrast, to the best of our knowledge, the proposed VT-FSL is the first to be extensively evaluated across three distinct FSL scenarios (standard, fine-grained, and cross-domain) on ten datasets, achieving a significant average improvement of 4.4% compared to existing SOTA methods.
>
> Furthermore, these methods are faithfully re-implemented in the missing scenarios by extensive hyperparameter tuning including batch size, learning rate, weight decay, and training epoch. The results are shown below:
>
> Cross‑domain miniImageNet → Places:
> | Method| 1-shot | 5-shot |
> |:-------|:----------------------|:----------------------|
> | C2-Net [78] | $56.78 ± 0.51$ | $75.38 ± 0.40$ |
> | SUITED [79] | $58.95 ± 0.45$ | $76.81 ± 0.38$ |
> | ours | $\\bf{73.68} ± 0.41$ | $\\bf{81.52} ± 0.33$ |
>
> Fine‑grained CUB:
> | Method| 1-shot | 5-shot |
> |:-------|:----------------------|:----------------------|
> | SemFew [27] | $84.49 ± 0.41$ | $93.82 ± 0.30$ |
> | ECER [26] | $85.95 ± 0.40$ | $93.13 ± 0.29$ |
> | ours | $\\bf{91.08} ± 0.28$ | $\\bf{94.63} ± 0.19$ |
>
> Even with carefully optimized implementations, VT-FSL consistently outperforms these methods (e.g., +5.1% in CUB and +14.7% in Places under the 1-shot setting), further confirming the effectiveness and generalization of VT-FSL framework in all three scenarios due to full utilization of complementary visual and textual prompts with on rich semantics.

---

> ### Author Response · Authors · 2025-08-06
> **Official Comment by Authors**
>
> Dear Reviewer QnTs,
>
> We hope this message finds you well. We are writing to kindly follow up on our rebuttal to check whether our responses have fully addressed your concerns. If so, we would greatly appreciate it if you would consider raising the final score. If you have any further concerns, please don't hesitate to discuss with us. We greatly value your feedback and will continue improving the final version based on the discussions.
>
> Thank you again for your time and thoughtful reviews.
>
> Best regards,
>
> Authors

---

### Note · Authors · 2025-08-12

We proposes VT-FSL, a framework that bridges vision and text for few-shot learning by constructing precise cross-modal prompts with large language models and integrating them via a geometry-aware alignment mechanism. The design combines Cross-modal Iterative Prompting, which generates visually grounded descriptions and semantically consistent synthetic images from both class names and support images. Cross-modal Geometric Alignment is then proposed to align textual, support, and synthetic visual representations simultaneously by minimizing the kernelized volume they form in a contrast learning manner.

Overall the reviewers appreciated the clear motivation, novel and timely idea with an interesting alignment mechanism. The experimental performance is superior and well-written presentation is easy to follow. Main concerns were raised about the computational cost of large models, the fairness and contribution of synthetic images, and whether improvements stem from the proposed modules rather than model scale or data volume.

These issues were addressed in detail through extensive theoretical and experimental analysis. First, in the proposed VT-FSL, large models are used only once in an offline prompt‑construction stage, without participating in downstream training. Training and inference times were compared with competitive methods, and the results show that VT-FSL achieves the highest accuracy with the lowest computational cost. Secondly, the generation of synthetic images strictly follows the protocol of few-shot learning (FSL) by using only given support samples without any other real data, which is consistency with prior FSL works using image generation and ensures FSL integrity. The contribution of synthetic images was also validated by confirming their strong visual consistency with the support set. Finally, ablations on backbone architectures, fusion factor, generation count, and individual textual/visual prompt terms verified that performance gains arise from the proposed modules and their complementary effects.

The reviewers were overall receptive to the rebuttal, and one explicitly improve the final rating as a result. This paper provides a novel generalized benchmark applying LLMs to few-shot learning by the extensive validation across three distinct scenarios on ten datasets. It also offers an effective multimodal integration method that is demonstrably better through comprehensive ablation studies.

---

### Decision · Program_Chairs · 2025-09-17

**Decision:**

Accept (poster)

**Comment:**

This work proposes a novel methodology to construct text prompts which are conditioned on Large Language Models and the support images in a few-shot learning scenario. To achieve this, authors integrate a geometry-aware mechanism that better aligns two modalities. During the review process, this work received mixed scores, with reviewers acknowledging its novelty, mathematical interpretability, and comprehensive and complete empirical validation. Nevertheless, they raised important concerns, mostly related to the fairness of the setting (i.e., encoder) used and the experimental analysis, a few inconsistent notations, and clarifications and suggestions to improve the readability of the manuscript. These weaknesses were mostly addressed during the post-rebuttal discussion process, where all the reviewers maintain a positive overview of this work, recommending borderline accept. Despite the rebuttal not fully addressing certain issues, the contributions and overall quality of the work are sufficient to justify its acceptance